# On Information Self-Locking in Reinforcement Learning
# for Active Reasoning of LLM agents

**Deyu Zou** [* 1]   **Yongqiang Chen** [* 1]   **Fan Feng** [2]   **Mufei Li** [3]   **Pan Li** [3]   **Yu Gong** [† 4]   **James Cheng** [‡ 1]

## Abstract

Reinforcement learning (RL) has become a *de facto* paradigm for building LLM-based agents that act, interact, and reason over extended task horizons. However, in ***active reasoning*** where agents must elicit new observations through interaction with the environment to solve the task, we find that outcome-based RL can induce a systematic failure mode which we call ***information self-locking (SeL)***: agents fail both to elicit informative feedback and to internalize obtained evidence. To understand the issue, we trace agentic behaviors into two coupled capabilities: ***Action Selection (AS)***, which determines observation streams, and ***Belief Tracking (BT)***, which updates the agent's internal task understanding. Theoretical and empirical analyses reveal a ***bidirectional*** bottleneck that leads to SeL: weak BT obscures the credit of informative actions, while weak AS deprives BT of useful evidence. This coupling weakens the learning signal for both capabilities and leads to SeL. To mitigate this issue, we propose AREW, a simple yet effective Advantage-Reweighting method that uses easy-to-obtain directional critiques to reallocate credit within trajectories. Extensive experiments across **9** agentic tasks of varying complexity show that AREW significantly mitigates SeL, yielding up to 60-point gains in final performance. Code is available at https://github.com/unimpor/T3.

## 1. Introduction

Reinforcement learning (RL) with outcome-based rewards has demonstrated great success in improving the reasoning capabilities of large language models (LLMs) (Wang et al., 2024; Srivastava & Aggarwal, 2025; Xu et al., 2025; Guo et al., 2025). Recently, this paradigm has become increasingly important for building LLM agents that solve tasks through extended interaction with external environments, rather than from a single prompt alone (Zhang et al., 2025a; Plaat et al., 2025). In contemporary agentic applications (Zhang et al., 2025b; Sager et al., 2026), such as deep research, coding, computer use, *etc.*, the agent often does not receive all task-relevant information upfront. Instead, the agent must make progress by actively taking environment-facing actions, receiving new observations or feedback, and updating its understanding of the task state over multiple turns. We refer to this setting as ***agentic active reasoning***.

Despite the success of outcome-based RL, we observe a recurring failure mode when training LLM-based agents for active reasoning. Rather than learning to acquire and use more task-relevant information, agents can drift into low-information interaction patterns: they take actions that reveal little useful evidence, and even when useful observations are returned, they do not reliably incorporate them into subsequent reasoning. We refer to this training regime as ***information self-locking*** (***SeL***). Under SeL, outcome-based RL may still improve apparent task performance through interaction-insensitive shortcuts, but it fails to strengthen the information acquisition and evidence-integration behaviors needed for active reasoning. This raises a crucial research question: *Why does SeL happen under outcome-based RL for agentic active reasoning, and how can we mitigate it?*

To answer this question, we start by exploring the coupled nature of the credit assignment system in active reasoning. Specifically, the agent's performance depends on two intertwined capabilities: ***Action Selection*** (***AS***), which determines the observation stream, and ***Belief Tracking*** (***BT***), which governs evidence internalization. Crucially, these two are mutually dependent: informative actions receive credit only when the resulting observations are properly utilized by BT, while effective BT refinement requires a high-quality information stream from AS. Concretely, empirical evidence (Sec. 2) reveals this coupling stagnation: even as rewards increase, both AS and BT capabilities remain dormant, suggesting that the sparse outcome reward fails to decouple and individually improve these two fundamental gears.

---

[*]Equal contribution  [†]Project lead  [‡]Corresponding author [1]The Chinese University of Hong Kong [2]University of California San Diego [3]Georgia Institute of Technology [4]ByteDance.

*Proceedings of the 43rd International Conference on Machine Learning*, Seoul, South Korea. PMLR 306, 2026. Copyright 2026 by the author(s).

*Figure 1.* Overall illustration of information self-locking (SeL) and its mitigation. $(b, a, o)$ denote the agent's internal belief, its chosen action, and the resulting feedback at each turn. Under vanilla outcome-based RL (top), the agent can become trapped in a self-locking regime: deficient belief tracking masks contributions of informative actions, leading to misaligned credit assignment. Our AREW (bottom) introduces advantage reweighting via directional critiques, correcting the learning signal and assisting to mitigate SeL in active reasoning.

To further formalize this intuition, we theoretically characterize the coupled learning dynamics of AS and BT under outcome-based optimization (Sec. 3). Our analysis decomposes the outcome-gradient into AS and BT channels and demonstrates that, in a low-information regime, their improvement signals become capability-limited: weak belief tracking reduces the reward sensitivity to informative actions, thereby masking the learning signal for AS; at the same time, deficient AS restricts the evidence stream available for belief refinement, limiting the upward drift of BT. Together, these effects form a coupled learning bottleneck (Thm. 3.4), where the outcome-gradient cannot provide a sufficiently strong improvement signal for either capability, causing the agent to remain trapped in a persistent low-information regime (SeL regime, Def. 3.3).

Built on this theoretical understanding, we propose AREW (Sec. 4), a lightweight and calibration-free framework for breaking SeL. The key observation is that active reasoning often provides easy-to-obtain local diagnostic signals, *e.g.*, whether a query is informative can be inferred from whether the user reveals new evidence in response. AREW uses them as weak *directional critiques* for AS and BT, rather than implementing calibrated intermediate rewards or training a dense reward model. It then injects these critiques at the policy-gradient level by reweighting stepwise advantages, reallocating credit within each trajectory from negatively critiqued decisions to positively critiqued ones while preserving the original outcome reward. This reward-preserving credit reallocation supplies non-degenerate learning signals in the SeL regime, enabling AS and BT to improve jointly and escape SeL.

Extensive experiments (Sec. 5) across **9** agentic active reasoning tasks of varying complexity show that AREW

consistently mitigates SeL across tasks, algorithms, and model families, and is robust to critique design of different noise levels. Beyond improving final performance, AREW fundamentally alters training dynamics: agents recover information-seeking interaction patterns and exhibit sustained growth in both action selection and belief tracking capabilities.

## 2. Self-Locking in Active Reasoning

### 2.1. Preliminaries

In active reasoning, an LLM agent solves a task under partial observability: the information provided at the beginning of an episode is insufficient for completing the task, and the agent must interact with an external environment to acquire task-relevant observations. At each turn, the agent takes an environment-facing action $a_t \in \mathcal{A}$, such as issuing a query or invoking a tool, and receives an observation $o_t \in \mathcal{O}$ from the environment. The agent must then use the accumulated interaction history to update its understanding of the task state and decide what action to take next.

We model active reasoning as a Partially Observable Markov Decision Process (POMDP) $(\mathcal{S}, \mathcal{A}, \mathcal{O}, T, O, R, \gamma)$ (Kaelbling et al., 1998), where $\mathcal{S}$ is the latent state space, $\mathcal{A}$ is the action space, $\mathcal{O}$ is the observation space, $T(s' \mid s, a)$ is the transition dynamics, $O(o \mid s, a)$ is the observation model, $R$ is the reward function, and $\gamma$ is the discount factor. In many agentic active reasoning tasks, the latent state $s^\star \in \mathcal{S}$ may represent the hidden user preference, diagnosis, solution, *etc*. The agent cannot directly observe $s^\star$. Belief tracking is central to active reasoning. For analysis, we associate the agent with a model belief $b_t \in \Delta(\mathcal{S})$ at turn $t \in \{0, \ldots, H\}$, which represents its internal understanding

of the latent task state and the current progress of problem solving. In this work, the belief is mainly leveraged as an analytical abstraction of the agent's task-state understanding induced by its parameters and interaction history.

Under this abstraction, the behavior of an agent with parameters $\omega$ can be decomposed into two coupled processes.

**Action Selection (AS).** Given its current belief, the agent selects an environment-facing action $a_t \sim \pi_\omega^{\mathrm{as}}(\cdot \mid b_t)$, which determines what observations may become available next through $o_t \sim O(\cdot \mid s^\star, a_t)$. The role of AS is therefore to shape the future observation stream by choosing actions that reduce uncertainty or otherwise advance task completion.

**Belief Tracking (BT).** After receiving the observation $o_t$, the agent updates its internal task-state understanding through a belief-update kernel $b_{t+1} \sim \pi_\omega^{\mathrm{bt}}(\cdot \mid b_t, a_t, o_t)$, which integrates the newly acquired observation with information accumulated from previous interaction rounds.

Together, these two kernels define the agent's interleaved behavior $\Pi_\omega := \left(\pi_\omega^{\mathrm{as}}, \pi_\omega^{\mathrm{bt}}\right)$. Over a horizon $H$, $\Pi_\omega$ induces trajectories $\tau = (b_0, a_0, o_0, b_1, \ldots, a_{H-1}, o_{H-1}, b_H)$ with likelihood $p_\omega(\tau)$ which can be factorized as follows,

$$p_\omega(b_0) \prod_{t=0}^{H-1} \pi_\omega^{\mathrm{as}}(a_t \mid b_t) O(o_t \mid s^\star, a_t) \pi_\omega^{\mathrm{bt}}(b_{t+1} \mid b_t, a_t, o_t).$$

This decomposition highlights the information pipeline underlying active reasoning: AS controls what evidence the agent can obtain, while BT controls whether that evidence is internalized and used for subsequent decisions.

## 2.2. Testbeds and decomposed proxies

With the above decomposition, we can track the fine-grained behaviors of agents trained with outcome-based RL. We now consider two interactive benchmarks, where we introduce proxies that separately track AS and BT dynamics. The full details can be found in Appendix C.1.

**Preference Estimation (PE).** Adapted from Badola et al. (2025), PE is an interactive preference inference task under constrained information acquisition. The agent is given a finite set of items $\mathcal{X} = \{x_1, \ldots, x_N\}$, where each item $x_i$ is represented by a known $D$-attribute vector $\mathbf{a}_i \in \mathbb{R}^D$. The user has an unknown latent preference vector $\mathbf{w}^\star \in [0, 1]^D$. Through interaction, the agent maintains and iteratively refines an estimate $\mathbf{w}_t \in [0, 1]^D$ of the user preference. At each round, the agent actively selects an attribute subspace $\mathcal{S}_t \subseteq \{1, \ldots, D\}$ where $|\mathcal{S}_t| = S$ and a pair of items $(x_i, x_j) \in \mathcal{X} \times \mathcal{X}$ designed to elicit the user's preference feedback restricted to $\mathcal{S}_t$. Based on the feedback, the agent updates its belief state. The objective is to recover $\mathbf{w}^\star$ under sparse, outcome-based supervision. We consider two variants, **PE-G**ated where $1 < S < D$, and **PE-F**ull where $S = D$, *i.e.,* all dimensions are covered each turn.

**Proxies in PE-G.** As it's hard to precisely quantify the informativeness, we introduce a binary proxy that is simple to implement and effective in tracking the AS behavior. Specifically, for a queried attribute subspace $\mathcal{S}_t$ and item pair $(x_i, x_j)$, we define the AS indicator $\mathrm{AS}_t = \mathbb{I}\left[\exists k_1, k_2 \in \mathcal{S}_t \text{ s.t., } a_i^{(k_1)} > a_j^{(k_1)} \wedge a_i^{(k_2)} < a_j^{(k_2)}\right]$, which means neither item strictly dominates the other on $\mathcal{S}_t$, ensuring that the resulting feedback is informative. BT evaluates whether the agent can incorporate such informative feedback. We measure BT by the improvement in similarity between the estimate and the ground-truth preference, $\mathrm{BT}_t = \mathrm{sim}(\mathbf{w}_{t+1}, \mathbf{w}^\star) - \mathrm{sim}(\mathbf{w}_t, \mathbf{w}^\star)$, where $\mathrm{sim}(\cdot, \cdot)$ denotes cosine similarity. Positive $\mathrm{BT}_t$ indicates effective absorption of newly acquired information.

**MediQ.** Adapted from Li et al. (2024), agents in MediQ require asking the patient questions to identify the best hypothesis for the patient's symptoms. The agent is provided with a clinical vignette and an associated medical question whose answer lies in a finite hypothesis set of size $D$. The agent maintains a belief estimate $\mathbf{w}_t \in [0, 1]^D$ for $D$ candidate hypotheses. Through interaction, the agent actively queries the LLM-simulated user for diagnostic information, receives structured feedback, and updates each hypothesis score accordingly. The learning objective is to progressively concentrate belief mass onto the correct hypothesis.

**Proxies in MediQ.** AS is quantified by the amount of novel diagnostic evidence elicited by the queries. Let $\mathcal{E}_t$ denote the set of atomic clinical facts revealed at turn $t$, we define $\mathrm{AS}_t = \left|\mathcal{E}_t \setminus \bigcup_{\tau < t} \mathcal{E}_\tau\right|$, to capture the information gain of each query. BT measures whether newly observed evidence sharpens hypothesis discrimination. Let $\mathrm{gt}$ denote the ground-truth hypothesis index. We define BT via the change in belief margin $\mathrm{BT}_t = \Delta\left(\mathbf{w}_t^{(\mathrm{gt})} - \max_{j \neq \mathrm{gt}} \mathbf{w}_t^{(j)}\right)$, aggregated across turns, where larger positive values indicate more effective belief refinement.

## 2.3. Failure modes in reinforcement learning training

Despite the success of RL with outcome-based rewards, interestingly, we find that LLM agents exhibit several failure modes across both active-reasoning testbeds during training.

**Observation 1: Reward improvements do not translate into increased information acquisition.** Fig. 2a (PE-G) and Fig. 2b (MediQ) report the training dynamics of episode reward, per-turn AS, and per-turn BT. Across both datasets, we observe a pronounced decoupling: while the reward can be improved over training, BT exhibits only limited gains, and AS fails to improve, often plateauing or even degrading. This observation raises interesting questions about the confounding behaviors of AS and BT, as AS can not improve even with an improved BT.

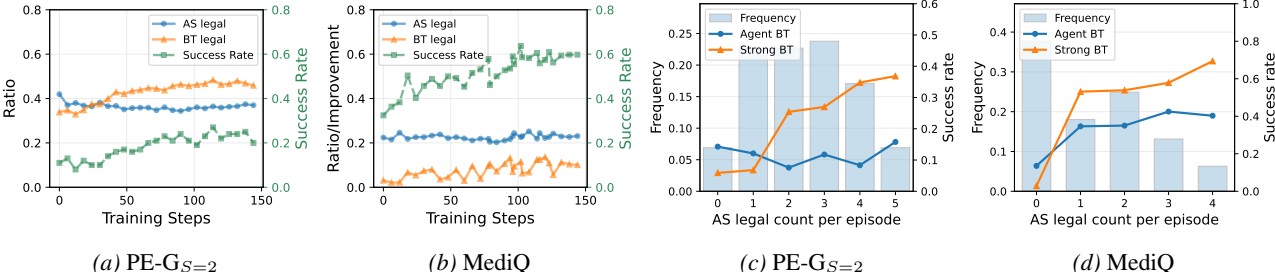

*Figure 2.* (a)/(b): the training dynamics of outcome reward, per-turn AS, and per-turn BT proxies in PE-G$_{S=2}$ and MediQ datasets (Qwen-2.5-7B-Instruct). (c)/(d): correlation between the reward and AS proxies in PE-G$_{S=2}$ and MediQ (Qwen-2.5-7B-Instruct). In strong BT patterns (by human-defined rules or frontier LLMs), the same AS sequence exhibits stronger correlation with the final reward.

To isolate the effect, we analyze the relationship between AS and reward under different BT capabilities. Specifically, we fix identical action sequences and compare outcomes when the observation stream is processed by (i) the agent's internal BT versus (ii) stronger belief-update mechanisms, *e.g.*, human-defined update rules or frontier reasoning models. Since the environment dynamics and action sequences are identical across conditions, the difference can be attributed solely to the belief update mechanism.

**Observation 2: Weak belief tracking masks the contribution of informative actions.** As shown in Fig. 2c and 2d, the correlation between AS and reward is substantially higher under strong BT, but remains weak when using the agent's own BT. This indicates that the contribution of AS to the reward is *masked* when belief updates are unreliable: even high-information actions yield little reward improvement if their information is not incorporated into internal belief. As a result, policy optimization cannot yield stable learning signals to reinforce informative AS choices.

**Observation 3: Conservative action selection limits belief refinement and induces interaction-insensitive shortcuts.** Complementary to Obs. 2, we now examine the reverse direction of the coupling. When AS gets conservative and yields little informative evidence, BT is deprived of meaningful signals to learn from. Under outcome-only supervision, this even incentivizes shortcut behaviors that reduce reliance on interaction, reinforcing a low-information training regime. We observe that as training progresses, agents become less sensitive to informative observations and increasingly rely on early-stage context. In MediQ, we intervene by replacing all patient feedback with `Unknown` while keeping all other configurations unchanged. Notably, the induced performance drop becomes smaller after RL training (41.25→30.50 w/o RL versus 61.00→55.50 with RL; see Fig. 6a), suggesting that interaction-derived evidence has a weaker causal effect on the final decision. Crucially, this reduced sensitivity is accompanied by an increase in *belief consistency* (Fig. 6a; 78.7 w/o RL versus 92.8 with RL): the agent increasingly adheres to its initial judgment instead of revising beliefs in response to interaction, which reflects a more "stubborn" belief update pattern. Together,

these form ***interaction under-utilization***: once conservative AS restricts information exposure and weak BT struggles to internalize evidence, RL pressure favors non-interactive heuristics that stabilize outcomes while further suppressing information exploration and evidence usage.

Taken together, these observations indicate SeL emerges from a bidirectional coupling between AS and BT. The reward-relevant value of AS is mediated by the agent's ability to absorb information through BT, while BT is in turn constrained by the information budget induced by AS. This mutual dependence can trap training dynamics in a low-information regime, giving rise to a self-locking behavior.

## 3. Understanding Self-Locking

To formally understand the SeL behaviors of AS and BT, we present a theoretical framework for SeL. Due to space constraints, we defer the full details to Appendix B.

**AS and BT capabilities.** We consider an agent represented by the interleaved behavior $\Pi_\omega = (\pi_\omega^{\mathrm{as}}, \pi_\omega^{\mathrm{bt}})$, where $\pi_\omega^{\mathrm{as}}$ selects environment-facing actions and $\pi_\omega^{\mathrm{bt}}$ updates the model belief. Let $\tau = (b_0, a_0, o_0, \ldots, b_H) \sim \Pi_\omega$ denote the on-policy trajectory, where $b_t$ is the model belief. To isolate AS from the model's BT mechanism, we also consider an oracle-belief trajectory $\bar{\tau} = (\bar{b}_0, \bar{a}_0, \bar{o}_0, \ldots, \bar{b}_H) \sim \Pi_\omega^{\mathrm{orc}}$, where barred variables denote **Bayesian belief updates** using the **same** action-selection kernel, and $\Pi_\omega^{\mathrm{orc}} := (\pi_\omega^{\mathrm{as}}, \mathsf{BayesUpd})$. We quantify belief quality by the potential $\Psi(b) := b(s^\star) \in [0, 1]$, which measures the confidence mass assigned to the true latent state $s^\star$. Then, we quantify AS and BT through the following two capability indices.

**Definition 3.1** (AS Informativeness). *Given an oracle-belief trajectory $\bar{\tau} \sim \Pi_\omega^{\mathrm{orc}}$ induced by the action-selection kernel $\pi_\omega^{\mathrm{as}}$ and Bayesian belief updates $\mathsf{BayesUpd}$, the AS informativeness is defined as the expected total improvement in oracle belief quality, $I_{\mathrm{AS}}(\omega) := \mathbb{E}_{\bar{\tau} \sim \Pi_\omega^{\mathrm{orc}}} \left[ \Psi(\bar{b}_H) - \Psi(\bar{b}_0) \right].$*

We next characterize how much of the theoretically supplied information is actually absorbed by the agent's belief-tracking dynamics.

**Definition 3.2** (BT Capability). *For an on-policy trajectory*

$\tau \sim \Pi_\omega$, *define* $\Delta\Psi_t := \Psi(b_{t+1}) - \Psi(b_t)$ *as the one-step model belief progress at turn* $t$, *and* $\Delta\Psi_t^+ := \max(\Delta\Psi_t, 0)$ *as the **absorbed** belief progress. The belief-tracking capability is defined as* $C_{\mathrm{BT}}(\omega) := \mathbb{E}_{\tau \sim \Pi_\omega}\left[\sum_{t=0}^{H-1} \Delta\Psi_t^+\right]$.

**Self-Locking regime.** We formalize the notion of self-locking via a two-dimensional low-AS and low-BT region:

**Definition 3.3** (Self-Locking Regime). *With* $\delta, \varepsilon > 0$, *we define the locking regime as the subset of parameter space with low AS informativeness and low BT capability:*

$$\mathcal{R}_{\delta,\varepsilon} := \left\{\omega \in \Omega : I_{\mathrm{AS}}(\omega) \leq \delta, \ C_{\mathrm{BT}}(\omega) \leq \varepsilon\right\},$$

*which represents a low-information and low-BT regime.*

**Policy-gradient decomposition of the outcome reward.** Given the agent $\Pi_\omega = (\pi_\omega^{\mathrm{as}}, \pi_\omega^{\mathrm{bt}})$ trained with outcome-based reward, the policy gradient can be written as

$$\nabla_\omega J(\omega) = \mathbb{E}_{\tau \sim \Pi_\omega}\left[R(\tau)\,\nabla_\omega \log p_\omega(\tau)\right].$$

Then we expand $\nabla_\omega \log p_\omega(\tau)$ as follows:

$$\sum_{t=0}^{H} \nabla_\omega \log \pi_\omega^{\mathrm{bt}}(b_t \mid c_t) + \sum_{t=0}^{H-1} \nabla_\omega \log \pi_\omega^{\mathrm{as}}(a_t \mid b_t).$$

Here $c_t$ denotes the BT-conditioning context. We analyze the **normalized** AS- and BT-channel update directions:

$$g_{\mathrm{as}}(\omega) = \mathbb{E}_\omega\left[\frac{1}{H}\sum_{t=0}^{H-1} \nabla_\omega \log \pi_\omega^{\mathrm{as}}(a_t \mid b_t) A_t^{\mathrm{as}}(b_t, a_t)\right],$$

$$g_{\mathrm{bt}}(\omega) = \mathbb{E}_\omega\left[\frac{1}{H+1}\sum_{t=0}^{H} \nabla_\omega \log \pi_\omega^{\mathrm{bt}}(b_t \mid c_t) A_t^{\mathrm{bt}}(c_t, b_t)\right],$$

where $A^{\mathrm{as}}$ and $A^{\mathrm{bt}}$ are the channel-isolated advantages of AS and BT, respectively. Naturally, we define the AS-channel projected update as $\mathcal{T}_{\mathrm{as}}(\omega) := \omega + \eta g_{\mathrm{as}}(\omega)$, and the BT-channel update as $\mathcal{T}_{\mathrm{bt}}(\omega) := \omega + \eta g_{\mathrm{bt}}(\omega)$. The coupling effects between AS and BT can be further characterized by the one-sided projected drifts:

$$\Delta_{\mathrm{as}}^+ I_{\mathrm{AS}}(\omega) := \left(I_{\mathrm{AS}}(\mathcal{T}_{\mathrm{as}}(\omega)) - I_{\mathrm{AS}}(\omega)\right)_+,$$

$$\Delta_{\mathrm{bt}}^+ C_{\mathrm{BT}}(\omega) := \left(C_{\mathrm{BT}}(\mathcal{T}_{\mathrm{bt}}(\omega)) - C_{\mathrm{BT}}(\omega)\right)_+.$$

With these quantities, we draw the following result:

**Theorem 3.4** (Informal). *Fix* $\delta, \varepsilon > 0$. *Under the regularity assumptions detailed in Appendix B.4, including Lipschitz reward-belief coupling, bounded gradients, action-invariant harmful belief drift, BT-capability-limited absorption, residual AS-budget comparability, and conservative propagation of belief quality, for any* $\omega \in \mathcal{R}_{\delta,\varepsilon}$, *the one-sided projected drifts satisfy the following inequality:*

$$\begin{pmatrix} \Delta_{\mathrm{as}}^+ I_{\mathrm{AS}}(\omega) \\ \Delta_{\mathrm{bt}}^+ C_{\mathrm{BT}}(\omega) \end{pmatrix} \preceq \eta \begin{pmatrix} 0 & \alpha \\ \beta_I & \beta_C \end{pmatrix} \begin{pmatrix} I_{\mathrm{AS}}(\omega) \\ C_{\mathrm{BT}}(\omega) \end{pmatrix} + o(\eta),$$

*where* $\preceq$ *denotes elementwise inequality and* $\alpha, \beta_I, \beta_C > 0$ *are problem-dependent constants. Moreover, for agents*

*initialized inside* $\mathcal{R}_{\delta,\varepsilon}$, *the agent cannot leave the locking regime for a non-trivial number of policy-update steps, with the explicit escape-time lower bound given in Appendix B.4.*

The formal version of Thm. 3.4 along with the proof are given in Appendix B.4. Intuitively, Thm. 3.4 shows that, when under the SeL regime, the learning signals from outcome reward are weakened by the limited AS and BT capabilities, scaling linearly with the current levels of $I_{\mathrm{AS}}$ and $C_{\mathrm{BT}}$. Consequently, when the model is initialized within the SeL regime, it requires significant policy update steps to escape the SeL regime. In practice, this indicates that once training enters the SeL regime, it is unlikely to recover without explicit interventions that restore informative local credit signals for AS and BT.

## 4. Breaking Self-Locking with Directions

The previous section shows that SeL arises from entangled and weakened credit assignment between AS and BT under outcome-based RL. While dense calibrated supervision for intermediate decisions is difficult in long-horizon agentic tasks, many active-reasoning environments provide easy-to-obtain **directional** signals at the step level; for example, an action's informativeness can often be inferred from whether the environment reveals new evidence in response. Motivated by this observation, we propose AREW, a lightweight framework that converts such uncalibrated directional signals into policy-gradient credit reallocation, improving AS and BT without introducing calibrated intermediate rewards.

### 4.1. Stepwise directional critiques

The AS/BT decomposition of agentic behavior allows us to exploit respective directional critiques. For analysis and implementation, the reasoning process of the agent can be organized as alternating between (i) an *Action Round*, in which the agent outputs an environment-facing action, and (ii) an *Update Round*, in which the agent receives the new observation and updates its internal belief.

**AS directional critique.** For the AS channel, we assign a directional critique $z_t^{\mathrm{as}} \in \{-1, 0, +1\}$ to each executed action, where $+1$ indicates that the action elicits informative feedback or useful evidence from the environment or user, $-1$ indicates an uninformative action, and $0$ denotes an abstention. Intuitively, $z_t^{\mathrm{as}}$ encourages agents to strategically take actions that induce information helpful to reasoning.

**BT directional critique.** For the BT channel, the critique should reflect whether newly acquired information is effectively incorporated into the agent's internal belief state. While we cannot directly access the agent's belief state, we can still acquire a scalar readout $\widehat{\Psi}_t \in [0, 1]$ that tracks task-relevant confidence over turns from the agent, such as through prompting. Importantly, $\widehat{\Psi}_t$ is used purely as

instrumentation: it is neither assumed to coincide with, nor to recover, the latent analytical belief $b_t$. We then define $z_t^{\mathrm{bt}} := \mathrm{Sign}\left(\widehat{\Psi}_{t+1} - \widehat{\Psi}_t\right)$, where positive values indicate that the readout moves in a truth-aligned direction.

### 4.2. Injecting directional critiques into policy-gradient

**Margin-aware auxiliary objective.** We inject directional critiques via an auxiliary objective that (i) acts locally at the critiqued steps and (ii) induces a gradient that can be combined with standard policy gradients without modifying task rewards. To this end, for a trajectory $\tau$ with labels $\{z_t\}$, define the positively and negatively critiqued index sets $\mathcal{P}_\tau := \{t \mid z_t = +1\}$, and $\mathcal{N}_\tau := \{t \mid z_t = -1\}$. Note that we use $z_t$ to denote $z_t^{\mathrm{as}}$, $z_t^{\mathrm{bt}}$, or both. The construction is channel-agnostic. Whenever counts $|\mathcal{P}_\tau| > 0$ and $|\mathcal{N}_\tau| > 0$, we define an ***intra-trajectory likelihood-margin*** objective

$$\widehat{\mathcal{L}}(\omega;\tau) := \frac{1}{|\mathcal{P}_\tau|}\sum_{t\in\mathcal{P}_\tau}\log\pi_{\omega,t} - \frac{1}{|\mathcal{N}_\tau|}\sum_{t\in\mathcal{N}_\tau}\log\pi_{\omega,t}. \quad (1)$$

Here $\log\pi_{\omega,t}$ denotes the aggregate log-probability assigned by the agent to the decision segment taken at step $t$ under parameters $\omega$. Note that the "decision" can correspond to an action-selection decision or a belief-update decision, and the construction here thus applies to both AS and BT channels.

Eq. 1 directly encourages the agent to increase the log-probability mass on positively critiqued decisions *relative to* negatively critiqued ones, without introducing calibrated intermediate rewards or training a separate discriminator for $z_t$. Crucially, Eq. 1 is additive over time and therefore naturally compatible with multi-turn credit assignment.

**Implied per-step coefficients.** Notably, Eq. 1 has a gradient of the same form as standard policy gradients:

$$\nabla_\omega\widehat{\mathcal{L}} = \sum_{t=0}^{H-1} u_t\nabla_\omega\log\pi_{\omega,t}; \quad u_t := \begin{cases} \frac{1}{|\mathcal{P}_\tau|} & \text{if } z_t = +1, \\ \frac{-1}{|\mathcal{N}_\tau|} & \text{if } z_t = -1, \\ 0 & \text{if } z_t = 0. \end{cases}$$

Here the sign of $u_t$ matches the critique direction, so the auxiliary gradient pushes probability mass in the intended direction. Furthermore, $\sum_{t=0}^{H-1} u_t = 0$ exhibits a centering property, which means the auxiliary term induces a *pure likelihood margin* rather than a uniform likelihood shift.

**Minimal injection via advantage reweighting.** Let $\mathcal{J}_{\mathrm{RL}}$ denote the standard actor surrogate used by a policy-gradient RL algorithm. We consider the augmented surrogate

$$\widehat{\mathcal{L}}_{\mathrm{aug}}(\omega) := \mathcal{J}_{\mathrm{RL}}(\omega) + \lambda\,\mathbb{E}_\tau\left[\widehat{\mathcal{L}}(\omega;\tau)\right], \quad (2)$$

where $\lambda > 0$ controls the strength of critique injection, and the expectation is taken over on-policy trajectories. The resulting gradient update can thus be written as

$$\nabla_\omega\widehat{\mathcal{L}}_{\mathrm{aug}}(\omega) \propto \mathbb{E}_\tau\left[\sum_{t=0}^{H-1}\left(A_t + \lambda\,u_t\right)\nabla_\omega\log\pi_{\omega,t}\right],$$

which shows that injecting the critiques requires only a *minimal* modification to the actor update: it suffices to apply an additive shaping to the advantage $\widehat{A}_t \leftarrow A_t + \lambda\,u_t$ while keeping the outcome reward, critic target, and the underlying RL optimization machinery unchanged. Since the coefficients $u_t$ are derived from the likelihood-margin objective, the resulting update can be directly interpreted as ***reallocating*** policy-gradient magnitude from negatively critiqued steps to positively critiqued ones within the same trajectory, aligned with the directional critiques.

### 4.3. Theoretical and Practical Discussion

In contrast to the theoretical characterization of SeL in Thm. 3.4, the directional critique breaks the negative confounding of limited AS and BT. Let $\widehat{\mathcal{T}}_{\mathrm{as}}(\omega)$ and $\widehat{\mathcal{T}}_{\mathrm{bt}}(\omega)$ denote the corresponding critique-shaped updates induced by the likelihood-margin objective in Eq. 2, respectively. The following proposition demonstrates that the effectiveness of AREW is characterized by the "weighted accuracy" of $z_t$ critiques. See the complete formulation in Appendix B.5.

**Proposition 4.1.** *Under the setting of Sec. 4.2 and Appendix B.5, denote the stepwise weight as* $w_t(\omega) := |u_t|\,\big|\bar{A}_t(\bar{b}_t, \bar{a}_t)\big|\,\|\nabla_\omega\log\pi_\omega^{\mathrm{as}}(\bar{a}_t \mid \bar{b}_t)\|^2$ *and let* $W(\omega) := \mathbb{E}\left[\sum_{t=0}^{H-1} w_t(\omega)\right]$. *Then the critique quality is measured by the weighted accuracy*

$$\mathrm{Acc}_{\mathrm{as}}(\omega) := \frac{\mathbb{E}\left[\sum_{t=0}^{H-1} w_t(\omega)\,\mathbf{1}\{z_t = y_t\}\right]}{\mathbb{E}\left[\sum_{t=0}^{H-1} w_t(\omega)\right]} \in [0,1].$$

*Moreover, the first-order improvement in AS informativeness induced by* AREW *satisfies (and the BT-side analogue)*

$$I_{\mathrm{AS}}\big(\widehat{\mathcal{T}}_{\mathrm{as}}(\omega)\big) - I_{\mathrm{AS}}\big(\mathcal{T}_{\mathrm{as}}(\omega)\big) = \eta\,W(\omega)\left(2\,\mathrm{Acc}_{\mathrm{as}}(\omega) - 1\right).$$

In particular, AREW does not rely on perfectly accurate critiques. The proposition shows that AREW is effective whenever $\mathrm{Acc}_{\mathrm{as}}(\omega) > \frac{1}{2}$; this only requires directional critiques to be better than random in weighted accuracy, not calibrated step rewards. We empirically show that AREW is robust to critique noise (see Robustness Analyses in Sec. 5.2).

**Practical discussion.** In practice, our framework naturally maps to LLM agents with *alternating* action-selection (AS) and belief-tracking (BT) rounds during interaction. In each AS round, the agent takes an environment-facing action intended to reduce the task uncertainty. In the subsequent BT round, the agent is instructed to explicitly update its confidence over the task states using the latest feedback, where the confidence assigned to the ground-truth candidate $s^\star$ induces $\widehat{\Psi}_t$ when training labels provide $s^\star$. In practice, simple critiques can assist to escape SeL and improve performance. For AS, action-level signals $z_t^{\mathrm{as}}$ can be directly inferred from user or environment feedback. For BT, stepwise critique labels $z_t^{\mathrm{bt}}$ can be constructed by compar-

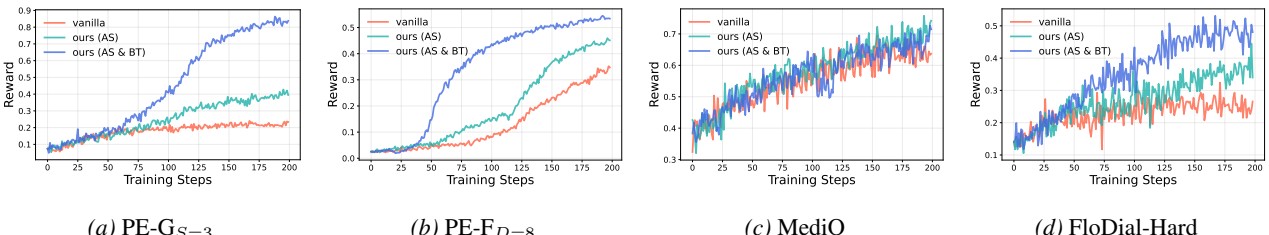

| | | | |
|:---:|:---:|:---:|:---:|
| *(a)* PE-G$_{S=3}$ | *(b)* PE-F$_{D=8}$ | *(c)* MediQ | *(d)* FloDial-Hard |

*Figure 3.* Training dynamics of rewards, evaluated under the PPO algorithm with Qwen-2.5-7B-Instruct across vanilla PPO, PPO with AREW− AS-ONLY and PPO with AREW− AS+BT.

*Table 1.* Main results (average outcome reward on test sets) across three domains and seven active reasoning tasks.

| | PREFERENCE ESTIMATION | | | | MEDICAL DIAGNOSIS | TROUBLESHOOTING | |
|---|---|---|---|---|---|---|---|
| | PE-G$_{S=2}$ | PE-G$_{S=3}$ | PE-F$_{D=8}$ | PE-F$_{D=6}$ | MEDIQ | FLODIAL-EASY | FLODIAL-HARD |
| **DIRECT INFERENCE** | | | | | | | |
| O4-MINI | 17.11 | 21.15 | 8.42 | 12.47 | 74.67 | 35.00 | 26.33 |
| QWEN-2.5-7B-INST. | 15.00 | 10.00 | 2.67 | 4.20 | 39.00 | 24.33 | 13.33 |
| LLAMA-3.1-8B-INST. | 18.00 | 12.33 | 3.14 | 5.70 | 35.25 | 24.33 | 17.33 |
| **PPO-TRAINED (QWEN-2.5-7B-INST.)** | | | | | | | |
| VANILLA | 24.00 | 18.33 | 30.52 | 32.03 | 50.50 | 37.33 | 21.33 |
| AREW− AS ONLY | 46.00 ↑22.0 | 32.00 ↑13.7 | 39.62 ↑9.1 | 42.10 ↑10.1 | 57.25 ↑6.8 | 43.67 ↑6.3 | 36.00 ↑14.7 |
| AREW− AS + BT | 49.33 ↑25.3 | 80.33 ↑62.0 | 47.89 ↑17.4 | 44.47 ↑12.4 | 61.25 ↑10.8 | 41.00 ↑3.7 | 42.33 ↑21.0 |
| **PPO-TRAINED (LLAMA-3.1-8B-INST.)** | | | | | | | |
| VANILLA | 27.33 | 11.00 | 55.21 | 6.00 | 63.50 | 24.33 | 31.00 |
| AREW− AS ONLY | 49.00 ↑21.7 | 73.00 ↑62.0 | 55.61 ↑0.4 | 56.91 ↑50.9 | 71.75 ↑8.3 | 41.67 ↑17.3 | 42.00 ↑11.0 |
| AREW− AS + BT | 54.67 ↑27.3 | 77.67 ↑66.7 | 61.28 ↑6.1 | 54.65 ↑48.7 | 70.75 ↑7.3 | 44.33 ↑20.0 | 49.00 ↑18.0 |

ing changes in candidate confidence (particularly that of $s^\star$) upon receiving feedback. We defer concrete prompt templates and implementation details to Appendix C.3.

## 5. Experiments

### 5.1. Experimental Setup

**Datasets.** We evaluate AREW across *4* interactive domains: preference estimation, medical diagnosis, troubleshooting, and customer service. Specifically, preference estimation includes PE-G and PE-F (Sec. 2); medical diagnosis corresponds to MediQ (Sec. 2); and troubleshooting is instantiated by FloDial in both easy and hard modes (Raghu et al., 2021; Hu et al., 2024), where an agent asks diagnostic questions to resolve user-reported issues. We further evaluate AREW on the customer-service domain using $\tau^2$-bench (Barres et al., 2025), a practical agentic benchmark in which the agent must interact with a user and external tools to complete realistic service tasks. Across all domains, supervision is provided only at the end of the interaction.

**Baselines.** To evaluate the effectiveness of AREW, we compare it against the following baselines: 1) Direct Inference without Training, where we evaluate representative proprietary reasoning LLMs o4-mini; 2) PPO (Schulman et al., 2017), 3) GRPO (Shao et al., 2024), and 4) GSPO (Zheng et al., 2025). See more details in Appendix C.2. For each RL mechanism, we consider two variants of AREW: AS-ONLY and AS+BT, which work on the AS side critique $z_t^{AS}$ and

both AS and BT sides $z_t^{AS}$, $z_t^{BT}$, respectively.

**Evaluation metrics.** We report average reward as well as AS and BT capability proxies on the test datasets. PE-F is evaluated under a continuous reward defined by normalized similarity improvement, whereas all other datasets use binary rewards. Details of rewards and the proxies used to approximate AS and BT are provided in Appendix C.3.

**Implementation details.** The main experiments of RL training are conducted on Qwen2.5-7B-Instruct (Yang et al., 2024) and LLaMA-3.1-8B-Instruct (Grattafiori et al., 2024). For the PE-G and PE-F tasks, the interactive feedback is rule-based; for the MediQ and FloDial datasets, we leverage Qwen2.5-14B-Instruct to simulate the "user" and provides the interactive feedback. See more details in Appendix C.3.

### 5.2. Experimental Results and Analyses

In this part, we present experimental results on the first three settings to evaluate the effectiveness of AREW and to analyze its impact on learning dynamics in agentic active reasoning. We begin with overall performance comparisons, followed by examining how AREW affects reward optimization, AS and BT capabilities, and robustness analysis.

**Overall Performance.** We first report the main results across the evaluated domains. Table 1 summarizes final-task performance. We further visualize dynamics of reward optimization and information-related behaviors. Specifically, Fig. 3 shows episode-level reward trajectories, while

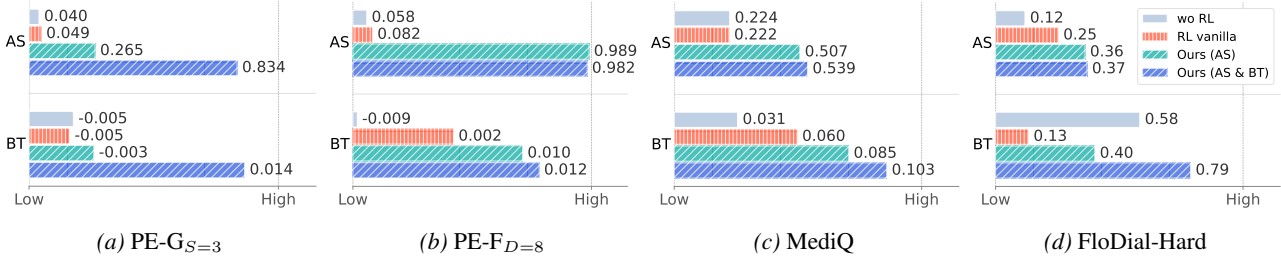

*(a)* PE-G$_{S=3}$     *(b)* PE-F$_{D=8}$     *(c)* MediQ     *(d)* FloDial-Hard

*Figure 4.* Evaluations of AS and BT capabilities under PPO algorithm with Qwen-2.5-7B-Instruct across vanilla PPO, PPO with AREW.

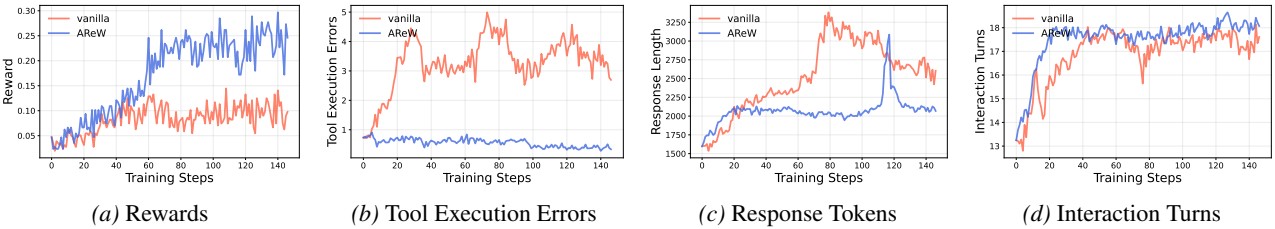

*(a)* Rewards     *(b)* Tool Execution Errors     *(c)* Response Tokens     *(d)* Interaction Turns

*Figure 5.* Training dynamics *wrt.* training steps on (a) rewards, (b) tool execution errors, (c) consumed response tokens and (d) interaction turns over Tau2Bench-Telecom solo setting, comparing vanilla PPO and PPO with AREW on Qwen2.5-7B-Instruct.

Fig. 4 reports proxy measurements of AS and BT capabilities introduced in Section 2. As shown in Table 1, AREW consistently outperforms the vanilla PPO baseline, achieving notable improvements in 27 out of 28 evaluated settings. Among these, AS+BT further largely outperforms the AS-ONLY variants in 11 out of 14 cases. These results indicate the effectiveness of AREW on mitigating self-locking.

**Reward dynamics.** Fig. 3 reveals three representative training behaviors. In some tasks (Fig. 3a), vanilla PPO fails to improve rewards throughout training, remaining trapped in SeL while AREW effectively breaks SeL and achieves continual improvement. In others (Fig. 3b and 3d), rewards can have some limited increase, but AREW achieves faster convergence and higher asymptotic performance. Notably, we also observe cases where reward curves appear comparable across methods, yet AREW yields superior AS and BT proxy scores and better final-task performance (See Table 1), corresponding to Obs. 3 discussed in Sec. 2.3.

**AS and BT behaviors.** Fig. 4 further disentangles AS and BT. The AS-ONLY variant outperforms the baseline in AS proxies in all cases, and the AS+BT variant achieves additional gains in BT over the AS-ONLY. Interestingly, the pure AS-ONLY variant already leads to measurable improvements in BT, reflecting the intrinsic coupling between information acquisition and belief updates, illustrating that breaking SeL brings benefits for both AS and BT channels.

**Effectiveness across different RL algorithms.** Beyond PPO, we additionally consider group-based RL variants, including GRPO and GSPO. While these methods sample multiple trajectories per step, we empirically observe that *self-locking behaviors can still arise* in interactive settings, suggesting that increasing rollout sampling alone may be insufficient to resolve the underlying coupling be-

*Table 2.* Final Performance under different strength of directional critique perturbation (controlled by $\alpha$).

| | VANILLA | PERTURBATION RATIO $\alpha$ | | | | | |
|---|---|---|---|---|---|---|---|
| | | 0 | 0.1 | 0.2 | 0.3 | 0.4 | 0.5 |
| PE-G$_{S=3}$ | 18.3 | 80.3 | 40.0 | 65.0 | 31.3 | 22.3 | 30.3 |
| FLODIAL-HARD | 21.3 | 36.0 | 30.3 | 29.0 | 27.6 | 30.6 | 23.3 |

tween AS and BT. As shown in Fig. 6b and 6c, AREW consistently improves final task performance, while simultaneously strengthening AS and BT proxies, analogous to those in PPO. These results indicate that AREW remains effective across different RL mechanisms.

**Robustness Analysis.** We evaluate the robustness of AREW by randomly flipping stepwise directional critiques with probability $\alpha$. As shown in Table 2, increasing the perturbation level leads to a gradual reduction in final performance. Nevertheless, AREW consistently remains competitive with, and often outperforms, the vanilla baseline across a wide range of $\alpha$. This trend is consistent with Proposition 4.1, which suggests that performance gains can be sustained as long as the weighted accuracy of critiques is not severely degraded. Even under strong perturbations (e.g., $\alpha = 0.5$), AREW does not collapse. Overall, these results indicate that AREW is robust to critique noise and can tolerate imperfect directional signals in practice.

### 5.3. Extension to Practical Customer-Service Agents

We further evaluate AREW on $\tau^2$-bench-Telecom (Barres et al., 2025), a practical customer-service benchmark that requires agents to solve telecom troubleshooting tasks through multi-turn dialogue and tool use. Unlike the controlled domains above, $\tau^2$-bench exposes a dual-control environment: the assistant can use backend tools, while the simulated user

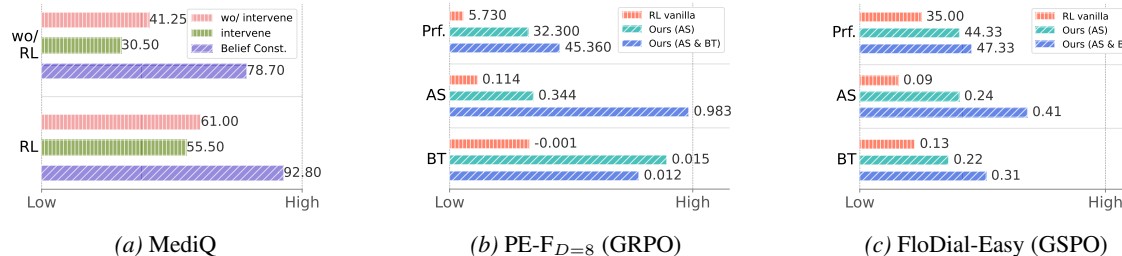

*(a)* MediQ     *(b)* PE-F$_{D=8}$ (GRPO)     *(c)* FloDial-Easy (GSPO)

*Figure 6.* (a): outcome-RL reduces sensitivity to interactive feedback while increasing belief consistency. (b)/(c): Evaluations of AS and BT capabilities under GRPO and GSPO (Qwen-2.5-7B-Instruct).

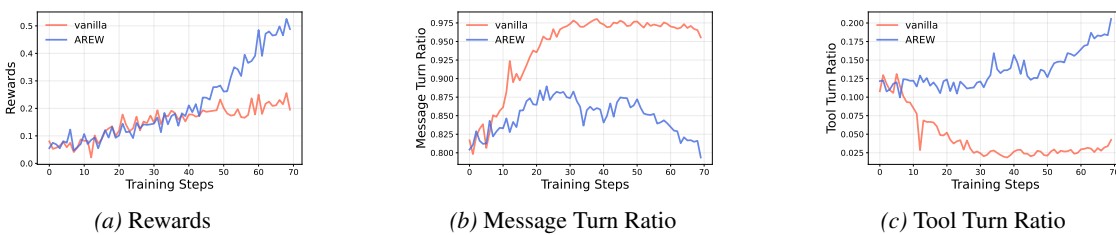

*(a)* Rewards     *(b)* Message Turn Ratio     *(c)* Tool Turn Ratio

*Figure 7.* Training dynamics on Tau2Bench-Telecom under the standard dual-control setting, comparing vanilla PPO and PPO with AREW on Qwen2.5-14B-Instruct. We report (a) rewards, (b) message-turn ratio, and (c) assistant tool-turn ratio.

can also execute user-side diagnostic or repair actions. This creates an additional credit-assignment challenge: the agent may obtain partial progress either by using its own tools correctly or by guiding the user to perform actions.

**Solo setting.** We consider the no-user solo setting, where the Qwen2.5-7B agent has direct control over the task-solving process. AREW is instantiated using only benchmark-native online signals, without additional annotation, preference data, or learned reward models. Negative critiques are derived from runtime failures such as malformed tool calls, invalid executions, and repeated actions, while positive critiques are derived from progress indicators exposed by the task evaluator, e.g., whether the trajectory newly matches an expected action. Fig. 5 shows that AREW substantially improves reward while sharply reducing tool execution errors. The improvement is not achieved by simply increasing interaction length or response verbosity: AREW uses fewer response tokens and maintains a comparable number of interaction turns. This suggests that lightweight directional critiques can provide useful optimization signals even in realistic long-horizon tool-use environments.

**Standard dual-control setting.** We study the standard interactive setting, where the Qwen2.5-14B agent must coordinate with a GPT-4o-simulated user while deciding when to communicate and when to use its own tools. This setting reveals a practical form of self-locking. Under vanilla PPO, early reward gains are largely associated with user-side operation: the policy increasingly guides the user to execute repair actions (Fig. 7b), while assistant-side tool use decreases (Fig. 7c). Although this behavior can solve a subset of tasks, it creates a path dependence toward the easiest available progress channel and weakens learning of

the intended assistant-side tool-use behavior. Trajectory inspection around step 60 further supports this interpretation: successful validation cases are dominated by user-side repair, but tasks requiring assistant-side repair remain hard.

AREW changes this learning dynamic. As shown in Fig. 7, over the matched 70-step training, AREW assigns more credit to useful assistant-side tool decisions and reduces over-reliance on user-side repair. Behaviorally, AREW maintains substantially more assistant read/write tool turns than vanilla PPO, while requiring fewer user tool hops, raising the average reward from $0.20$ to $0.50$, a $2.5\times$ relative increase over vanilla PPO. Thus, in $\tau^2$-bench, AREW not only improves early reward but also counteracts an interaction-induced shortcut, reallocating optimization pressure toward the agent-side behavior that the benchmark is designed to evaluate.

## 6. Conclusion

We study *information self-locking* (SeL) in long-horizon multi-turn agentic active reasoning and show that it arises from a structural failure of credit assignment with bidirectional coupling between action selection (AS) and belief tracking (BT). We provide both theoretical and empirical evidence that standard outcome-based RL can be trapped in SeL. We propose AREW, a critique-driven reweighting approach that selectively reallocates optimization signal along trajectories. Experiments demonstrate consistent gains, robustness to noisy critiques, effectiveness on multiple RL mechanisms, and improved training dynamics, AS and BT capabilities across multiple benchmarks. We believe this perspective opens up new directions for designing robust learning mechanisms for interactive reasoning agents.

## Acknowledgements

We thank the reviewers for their constructive comments and suggestions. Deyu Zou, Yongqiang Chen, and James Cheng were supported by a CRF (No. C2005-24Y) from the RGC of Hong Kong. Mufei Li and Pan Li are in part supported by NSF grants III-2239565 and III-2428777. This work was a collaboration between Husky Data+AI Lab at CUHK and ByteDance, supported by a ByteDance University Collaboration Project.

## Impact Statement

This paper presents work whose goal is to advance the field of Machine Learning. There are many potential societal consequences of our work, none which we feel must be specifically highlighted here.

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

# A. Related Work

**Active Reasoning** requires LLMs to interact with external sources and actively acquire missing information to solve complex tasks. Prior work has improved LLMs' ability to handle ambiguity and incompleteness through making clarification and information-seeking actions. For example, Proactive CoT (Deng et al., 2023) prompts LLMs to identify ambiguous problems and generate clarification questions, while UoT (Hu et al., 2024) quantifies the contribution of each question in reducing uncertainty. However, challenges remain when transitioning from LLMs' single-turn success to multi-turn active reasoning (Kwan et al., 2024; Liang et al., 2024; Badola et al., 2025), even with several advanced strategies such as tree-based searching or post-training approaches, as highlighted in existing works (Zhou et al., 2025). There have been existing works which target RL on active reasoning (Zou et al., 2026). In our work, we identify a unique mechanism named information self-locking, which is sourced from a structural failure of credit assignment with bidirectional coupling between action selection (AS) and belief tracking (BT). This is consistent with empirical observations from real-world agentic use. For example, Wang et al. (2025b) uncovers the phenomenon specific to search agents: systematic deficiencies in search behaviors arise under outcome-only training and ultimately degrade final answer quality.

**Credit Assignment and Multi-turn RL.** Credit assignment is crucial to long-horizon or multi-turn RL. Existing methods have extensively explored rule-based approaches (Yu et al., 2024; Dou et al., 2024; Zhang et al., 2025c) to shape intermediate rewards. Several recent works also proposed to measure the progress of stepwise actions toward overall task completion as intermediate rewards. Specifically, CURIO (Wan et al., 2025) constructs a potential function over an ideal belief state to assign intermediate rewards, assuming that the latent state space is finite and enumerable. Sotopia-RL (Yu et al., 2025) relies on reward labeling with proprietary LLMs. SPA-RL (Wang et al., 2025a) trains reward models for intermediate rewards by enforcing a summation constraint with respect to the final outcome reward. In our work, instead of working on complicated reward shaping or resorting to external models, we leverage easy-to-use binary directional critiques to make a minimal injection to policy gradient which is mathematically derived from a margin-aware auxiliary objective, in order to provide non-degenerate and stable learning signals to help agents escape from self-locking.

# B. More Details on the Theory

## B.1. Notations & Problem Setup

We consider *active reasoning*, where an LLM agent interacts with an external environment to acquire missing information and infer the underlying solution through multi-turn interaction. This can be modeled as a POMDP $(\mathcal{S}, \mathcal{A}, \mathcal{O}, T, O, R, \gamma)$, where $\mathcal{S}$ is the space of unobservable latent states, $\mathcal{A}$ the action space, $\mathcal{O}$ the observation space, $T(s' \mid s, a)$ the transition dynamics, $O(o \mid s, a)$ the observation model, $R$ the reward function, and $\gamma$ the discount factor. In our setting, the latent state is fixed within each episode of horizon $H$, denoted by $s^\star \in \mathcal{S}$. The action $a_t$ denotes a generic environment-facing action, such as issuing a query, retrieving information, invoking a tool, or executing an operation. We assume deterministic feedback: for all $(s, a) \in \mathcal{S} \times \mathcal{A}$, the observation kernel $O(\cdot \mid s, a)$ is a point mass at $o = \mathsf{O}(s, a)$; in particular, $o_t = \mathsf{O}(s^\star, a_t)$.

**Model Belief.** We work with a belief-state abstraction where the agent is associated with an explicit *model belief* $b_t \in \Delta(\mathcal{S})$, which represents the agent's internal understanding of the latent state and what information remains missing at each turn $t \in \{0, \ldots, H\}$. Note that this abstraction does *not* assume the agent internally stores $b_t$ as an explicit probability vector. $b_t$ is a well-defined analytical object induced by the model parameters $\omega$ and the interaction history. We quantify truth-aligned confidence by the potential $\Psi(b) := b(s^\star) \in [0, 1]$, which measures the belief mass assigned to the true latent state and serves as our working notion of belief quality.

**Model Trajectory.** Agentic behavior in active reasoning can be decomposed into two coupled processes: **Action Selection (AS)** and **Belief Tracking (BT)**. An agent with parameters $\omega$ induces: (i) an action-selection kernel $\pi_\omega^{\mathrm{as}}$ that selects environment-facing actions conditioned on the current belief, and (ii) a belief-update kernel $\pi_\omega^{\mathrm{bt}}$ that produces the next belief given the latest interaction. We denote the resulting interleaved agent behavior by $\Pi_\omega := (\pi_\omega^{\mathrm{as}}, \pi_\omega^{\mathrm{bt}})$. Concretely, define the initial BT-conditioning context $c_0$ as a fixed initial context (e.g., task instructions), and $c_{t+1} := (b_t, a_t, o_t)$ for $t \in \{0, \ldots, H-1\}$. For each turn $t$, the induced belief process evolves through $a_t \sim \pi_\omega^{\mathrm{as}}(\cdot \mid b_t)$, $o_t \sim O(\cdot \mid s^\star, a_t)$, and $b_{t+1} \sim \pi_\omega^{\mathrm{bt}}(\cdot \mid c_{t+1})$. We write the full trajectory induced by $\Pi_\omega$ as $\tau = (b_0, a_0, o_0, \ b_1, a_1, o_1, \ \ldots, \ b_{H-1}, a_{H-1}, o_{H-1}, \ b_H)$. Conditioned on $s^\star$ and omitting the fixed task context from notation, its likelihood factorizes as

$$p_\omega(\tau \mid s^\star) = \pi_\omega^{\mathrm{bt}}(b_0 \mid c_0) \prod_{t=0}^{H-1} \pi_\omega^{\mathrm{as}}(a_t \mid b_t) \, O(o_t \mid s^\star, a_t) \, \pi_\omega^{\mathrm{bt}}(b_{t+1} \mid c_{t+1}), \quad c_{t+1} = (b_t, a_t, o_t). \tag{3}$$

The environment factor $O(o_t \mid s^\star, a_t)$ is independent of $\omega$ and can be omitted when deriving policy-gradient updates.

**Outcome reward.** The environment returns an outcome reward $R(\tau) \in [0, 1]$ (e.g., correctness of the inferred solution). We assume the expected reward is a non-decreasing function of the terminal belief quality:

**Assumption B.1.** *There exists a non-decreasing $L_R$-Lipschitz function $f : [0, 1] \to [0, 1]$ such that $\mathbb{E}[R(\tau) \mid b_H] = f(\Psi(b_H))$.*

## B.2. Two Capability Indices

**Oracle Bayesian belief.** To decouple AS informativeness from the model's BT mechanism, we introduce an oracle Bayesian belief-update process. Given deterministic feedback $o = \mathsf{O}(s^\star, a)$, define the Bayesian update operator $\mathsf{BayesUpd}(\cdot, a, o)$:

$$\big(\mathsf{BayesUpd}(b, a, o)\big)(s) := \frac{b(s)\, \mathbf{1}\{\mathsf{O}(s, a) = o\}}{\sum_{s' \in \mathcal{S}} b(s')\, \mathbf{1}\{\mathsf{O}(s', a) = o\}}.$$

**Oracle-belief process.** For analysis purposes only, we consider an oracle-belief process that evolves under Bayesian belief updates while sharing the same action-selection kernel as the agent. Fix a prior $\bar{b}_0 \in \Delta(\mathcal{S})$ with $\bar{b}_0(s^\star) > 0$. For each turn $t \in \{0, \ldots, H - 1\}$, the oracle trajectory evolves as $\bar{a}_t \sim \pi_\omega^{\mathrm{as}}(\cdot \mid \bar{b}_t)$, $\bar{o}_t = \mathsf{O}(s^\star, \bar{a}_t)$, and $\bar{b}_{t+1} = \mathsf{BayesUpd}(\bar{b}_t, \bar{a}_t, \bar{o}_t)$. This yields the oracle trajectory $\bar{\tau} := (\bar{b}_0, \bar{a}_0, \bar{o}_0, \bar{b}_1, \bar{a}_1, \bar{o}_1, \ldots, \bar{b}_H) \sim \Pi_\omega^{\mathrm{orc}}$, where $\Pi_\omega^{\mathrm{orc}} := (\pi_\omega^{\mathrm{as}}, \mathsf{BayesUpd})$. Here, barred variables are used exclusively for this oracle process. The oracle-belief process is introduced solely for analysis, aiming to isolate AS information supply, and does not require the actual agent to access the oracle belief $\bar{b}_t$.

We now quantify the informativeness of the action-selection kernel by the belief improvement it induces under the oracle-belief dynamics.

**Definition B.2.** *Consider an oracle-belief trajectory $\bar{\tau} := (\bar{b}_0, \bar{a}_0, \bar{o}_0, \ldots, \bar{b}_H) \sim \Pi_\omega^{\mathrm{orc}}$ induced by the AS kernel $\pi_\omega^{\mathrm{as}}$ and Bayesian belief dynamics $\mathsf{BayesUpd}(\cdot)$. Define the one-step oracle belief progress as $\Delta\bar{\Psi}_t := \Psi(\bar{b}_{t+1}) - \Psi(\bar{b}_t)$. The AS informativeness of $\pi_\omega^{\mathrm{as}}$ is defined as the expected total improvement in oracle belief quality,*

$$I_{\mathrm{AS}}(\omega) := \mathbb{E}_{\bar{\tau} \sim \Pi_\omega^{\mathrm{orc}}} \left[ \sum_{t=0}^{H-1} \Delta\bar{\Psi}_t \right] = \mathbb{E}_{\bar{\tau} \sim \Pi_\omega^{\mathrm{orc}}} \left[ \Psi(\bar{b}_H) - \Psi(\bar{b}_0) \right].$$

We next characterize how much of the theoretically supplied information is actually absorbed by the agent's belief-tracking dynamics.

**Definition B.3.** *For an on-policy trajectory $\tau = (b_0, a_0, o_0, \ldots, b_H) \sim \Pi_\omega$, define the one-step model belief progress at turn $t$ as $\Delta\Psi_t := \Psi(b_{t+1}) - \Psi(b_t)$. We decompose $\Delta\Psi_t$ into its one-sided components: the absorbed belief progress $\Delta\Psi_t^+ := (\Delta\Psi_t)_+$, and the self-destructive belief drift $\Delta\Psi_t^- := (-\Delta\Psi_t)_+$. Only the absorbed component contributes positively to belief improvement. The training-level belief-tracking (BT) index is defined as*

$$C_{\mathrm{BT}}(\omega) := \mathbb{E}_{\tau \sim \Pi_\omega} \left[ \sum_{t=0}^{H-1} \Delta\Psi_t^+ \right].$$

To relate AS informativeness to BT progress under learning dynamics, we impose the following mild regularity assumptions. Each assumption is accompanied by a brief intuition clarifying its role: **Asmp. B.4** states that harmful belief drift is independent of the specific action chosen, and is instead driven by deficiencies in the agent's BT mechanism. **Asmp. B.5** upper-bounds the amount of belief progress that can be absorbed in a single step by the current BT capability. **Asmp. B.6** states that the residual oracle AS information budget from any reachable belief remains controlled by the global AS informativeness, reflecting that information supply is governed by the action-selection kernel itself, and no reachable belief state should unlock a much larger information budget than the global oracle-reference AS $I_{\mathrm{AS}}$. **Asmp. B.7** enforces a Lipschitz-type stability of future belief quality with respect to the current potential, reflecting a conservative propagation of belief improvements under weak belief tracking.

**Assumption B.4.** *For any turn $t$ and action $a \in \mathcal{A}$, the conditional expectation of the self-destructive belief drift given the current belief is invariant to the action choice, i.e., $\mathbb{E}_\omega\big[\Delta\Psi_t^- \mid b_t, a_t = a\big] = \mathbb{E}_\omega\big[\Delta\Psi_t^- \mid b_t\big]$ almost surely for all $a \in \mathcal{A}$.*

**Assumption B.5.** *For any turn $t$, there exists a constant $\kappa_{\mathrm{bt}} \geq 1$ such that along on-policy rollouts, the conditional expectation satisfies $\mathbb{E}_\omega\big[\Delta\Psi_t^+ \mid b_t, a_t\big] \leq \kappa_{\mathrm{bt}} C_{\mathrm{BT}}(\omega)$ almost surely over the realized action $a_t$.*

**Assumption B.6.** *For any turn $t$ and any reachable belief $b \in \Delta(\mathcal{S})$ considered in the SeL analysis, let $V_{\mathrm{AS},t}^{\mathrm{orc}}(b) := \mathbb{E}_{\bar{\tau}_{t:H} \sim \Pi_\omega^{\mathrm{orc}}(\cdot | \bar{b}_t = b)} \big[ \Psi(\bar{b}_H) - \Psi(b) \big]$ denote the residual oracle AS information budget from $b$ under the same action-selection kernel. There exists a constant $\kappa_{\mathrm{or}} > 0$ such that $V_{\mathrm{AS},t}^{\mathrm{orc}}(b) \leq \kappa_{\mathrm{or}} I_{\mathrm{AS}}(\omega)$ almost surely over such reachable beliefs.*

**Assumption B.7.** *For any $0 < t_0 < t_1 \leq H$, there exists a constant $\kappa_{\mathrm{pr}}$ such that for any realizable model beliefs $b, b' \in \Delta(\mathcal{S})$ at turn $t_0$, $|\mathbb{E}_\omega[\Psi(b_{t_1}) \mid b_{t_0} = b] - \mathbb{E}_\omega[\Psi(b_{t_1}) \mid b_{t_0} = b']| \leq \kappa_{\mathrm{pr}} |\Psi(b) - \Psi(b')|.$*

**Locking regime.** We formalize self-locking as a two-dimensional region with simultaneously low AS informativeness and low BT capability:

**Definition B.8** (Locking regime)**.** *For given thresholds $\delta, \varepsilon > 0$, we define the* locking regime *as the subset of parameter space characterized by simultaneously low action-selection informativeness and low belief-tracking capability:*

$$\mathcal{R}_{\delta,\varepsilon} := \big\{ \omega \in \Omega : I_{\mathrm{AS}}(\omega) \leq \delta, \ C_{\mathrm{BT}}(\omega) \leq \varepsilon \big\}.$$

*This region captures a low-information and low-BT regime in which neither the action-selection kernel nor the belief-tracking dynamics can induce substantial positive progress.*

To analyze local training dynamics within the locking regime, we require a mild first-order regularity condition on the two capability indices.

**Assumption B.9** (Local first-order expandability of capability indices)**.** *Fix $\delta, \varepsilon > 0$ and consider $\omega \in \mathcal{R}_{\delta,\varepsilon}$. There exist constants $M_{\mathrm{AS}}, M_{\mathrm{BT}} < \infty$ such that $\|\nabla_\omega I_{\mathrm{AS}}(\omega)\| \leq M_{\mathrm{AS}}$ and $\|\nabla_\omega C_{\mathrm{BT}}(\omega)\| \leq M_{\mathrm{BT}}$. Moreover, for any $\omega' \in \mathcal{R}_{\delta,\varepsilon}$ satisfying $\|\omega' - \omega\| = O(\eta)$, the following first-order expansions hold uniformly over $\mathcal{R}_{\delta,\varepsilon}$: $I_{\mathrm{AS}}(\omega') = I_{\mathrm{AS}}(\omega) + \langle \nabla_\omega I_{\mathrm{AS}}(\omega), \ \omega' - \omega \rangle + o(\eta)$, and $C_{\mathrm{BT}}(\omega') = C_{\mathrm{BT}}(\omega) + \langle \nabla_\omega C_{\mathrm{BT}}(\omega), \ \omega' - \omega \rangle + o(\eta)$, where $o(\eta)/\eta \to 0$ as $\eta \to 0$ uniformly for $\omega \in \mathcal{R}_{\delta,\varepsilon}$.*

### B.3. Projected Drifts and the 2D Locking Regime

**Policy-gradient decomposition of the outcome objective.** We begin by expressing the outcome-based policy gradient in a form that admits a decomposition across the AS and BT channels. With the outcome objective $J(\omega) := \mathbb{E}_{\tau \sim \Pi_\omega}[R(\tau)]$,

$$\nabla_\omega J(\omega) = \mathbb{E}_{\tau \sim \Pi_\omega} \Big[ R(\tau) \nabla_\omega \log p_\omega(\tau) \Big],$$

and hence (via Eq. 3, omitting the environment factors independent of $\omega$)

$$\nabla_\omega \log p_\omega(\tau) = \sum_{t=0}^{H} \nabla_\omega \log \pi_\omega^{\mathrm{bt}}(b_t \mid c_t) + \sum_{t=0}^{H-1} \nabla_\omega \log \pi_\omega^{\mathrm{as}}(a_t \mid b_t).$$

We next impose a mild regularity assumption to control the score-function magnitudes.

**Assumption B.10.** *There exist finite constants $G_{\mathrm{as}} < \infty$ and $G_{\mathrm{bt}} < \infty$ such that for all $\omega \in \mathcal{R}_{\delta,\varepsilon}$, we have $\|\nabla_\omega \log \pi_\omega^{\mathrm{as}}(a_t \mid b_t)\| \leq G_{\mathrm{as}}$ for all $t \in \{0, \dots, H-1\}$ and $\|\nabla_\omega \log \pi_\omega^{\mathrm{bt}}(b_t \mid c_t)\| \leq G_{\mathrm{bt}}$ for all $t \in \{0, \dots, H\}$ almost surely under $\tau \sim \Pi_\omega$.*

**Channel-isolated stage-wise advantages.** As mentioned before, we have decomposed the agentic behavior into two coupled but conceptually distinct channels: AS and BT. To characterize how outcome-based training signals propagate through each channel, we introduce channel-isolated stage-wise value functions with respect to the outcome objective $J$, where the other channel is treated as *part of the induced environment dynamics*. All expectations below use the continuation dynamics induced by $\Pi_\omega$ after conditioning on the specified channel decision.

For the BT channel, the belief state $b_t$ itself is regarded as the decision variable. For $t \in \{0, \dots, H\}$, define

$$Q_t^{\mathrm{bt}}(c, b) := \mathbb{E}[R(\tau) \mid c_t = c, \ b_t = b], \qquad V_t^{\mathrm{bt}}(c) := \mathbb{E}_{b \sim \pi_\omega^{\mathrm{bt}}(\cdot | c)} \big[ Q_t^{\mathrm{bt}}(c, b) \big],$$

and the corresponding advantage

$$A_t^{\mathrm{bt}}(c, b) := Q_t^{\mathrm{bt}}(c, b) - V_t^{\mathrm{bt}}(c).$$

Similarly, for the AS channel, the action $a_t$ is treated as the decision variable. For $t \in \{0, \dots, H-1\}$, define

$$Q_t^{\mathrm{as}}(b, a) := \mathbb{E}[R(\tau) \mid b_t = b, \ a_t = a], \qquad V_t^{\mathrm{as}}(b) := \mathbb{E}_{a \sim \pi_\omega^{\mathrm{as}}(\cdot | b)}[Q_t^{\mathrm{as}}(b, a)],$$

with advantage

$$A_t^{\mathrm{as}}(b, a) := Q_t^{\mathrm{as}}(b, a) - V_t^{\mathrm{as}}(b).$$

**Channel-isolated outcome update directions.** Using the above definitions, the raw channel-wise policy-gradient components are sums over the corresponding decisions. For scale stability across horizons, we analyze their normalized versions:

$$g_{\mathrm{as}}(\omega) := \mathbb{E}_{\tau \sim \Pi_\omega} \left[ \frac{1}{H} \sum_{t=0}^{H-1} \nabla_\omega \log \pi_\omega^{\mathrm{as}}(a_t \mid b_t) \, A_t^{\mathrm{as}}(b_t, a_t) \right],$$

$$g_{\mathrm{bt}}(\omega) := \mathbb{E}_{\tau \sim \Pi_\omega} \left[ \frac{1}{H+1} \sum_{t=0}^{H} \nabla_\omega \log \pi_\omega^{\mathrm{bt}}(b_t \mid c_t) \, A_t^{\mathrm{bt}}(c_t, b_t) \right].$$

Accordingly, define the AS-channel projected update $\mathcal{T}_{\mathrm{as}}(\omega) := \omega + \eta \, g_{\mathrm{as}}(\omega)$, and the BT-channel projected update $\mathcal{T}_{\mathrm{bt}}(\omega) := \omega + \eta \, g_{\mathrm{bt}}(\omega)$. These are *virtual* updates used only for mechanism analysis.

### B.4. 2D One-Sided Self-Locking via Projected Drifts

We begin by bounding the channel-isolated outcome advantages, which quantify the strength of the outcome-based learning signal available to each channel.

**Proposition B.11** (AS-channel outcome advantages). *Under Assumptions B.1, B.4, B.5, and B.7, for any on-policy executed action $a_t$ at turn $t$, $\mathbb{E}_\omega\big[\big|A_t^{\mathrm{as}}(b_t, a_t)\big|\big] \leq L_R \, \kappa_{\mathrm{pr}} \, \kappa_{\mathrm{bt}} \, C_{\mathrm{BT}}(\omega)$. Moreover, under Assumption B.10, $\|g_{\mathrm{as}}(\omega)\| \leq K_{\mathrm{as}} C_{\mathrm{BT}}(\omega)$, where $K_{\mathrm{as}} := G_{\mathrm{as}} \, L_R \, \kappa_{\mathrm{pr}} \, \kappa_{\mathrm{bt}}$.*

**Proposition B.12** (BT-channel outcome advantages). *Assume Assumptions B.1, B.4, B.5, and B.7 hold, then for any on-policy generated belief $b_t$ at turn $t$, $\mathbb{E}_\omega\big[\big|A_t^{\mathrm{bt}}(c_t, b_t)\big|\big] \leq 2L_R\big(C_{\mathrm{BT}}(\omega) + \kappa_{\mathrm{or}} \, I_{\mathrm{AS}}(\omega)\big)$. Moreover, under Assumption B.10, $\|g_{\mathrm{bt}}(\omega)\| \leq K_{\mathrm{bt},I} \, I_{\mathrm{AS}}(\omega) + K_{\mathrm{bt},C} \, C_{\mathrm{BT}}(\omega)$, where $K_{\mathrm{bt},I} := 2G_{\mathrm{bt}} \, L_R \kappa_{\mathrm{or}}$ and $K_{\mathrm{bt},C} := 2G_{\mathrm{bt}} \, L_R$.*

We now translate the channel-wise gradient bounds into one-step projected drifts of the two capability indices within the locking regime. For convenience, we denote by $\Delta_{\mathrm{as}}^+$ and $\Delta_{\mathrm{bt}}^+$ the positive parts of the one-step changes induced by the projected updates $\mathcal{T}_{\mathrm{as}}$ and $\mathcal{T}_{\mathrm{bt}}$.

**Proposition B.13** (AS projected drift is controlled by BT level). *Under Assumption B.9 and the conclusion of Proposition B.11, for all $\omega \in \mathcal{R}_{\delta,\varepsilon}$,*

$$\Delta_{\mathrm{as}}^+ I_{\mathrm{AS}}(\omega) := \big(I_{\mathrm{AS}}(\mathcal{T}_{\mathrm{as}}(\omega)) - I_{\mathrm{AS}}(\omega)\big)_+ \leq \eta \, \alpha \, C_{\mathrm{BT}}(\omega) + o(\eta), \qquad \alpha := M_{\mathrm{AS}} K_{\mathrm{as}}.$$

**Proposition B.14** (BT projected drift is controlled by AS and BT levels). *Under Assumption B.9 and the conclusion of Proposition B.12, for all $\omega \in \mathcal{R}_{\delta,\varepsilon}$,*

$$\Delta_{\mathrm{bt}}^+ C_{\mathrm{BT}}(\omega) := \big(C_{\mathrm{BT}}(\mathcal{T}_{\mathrm{bt}}(\omega)) - C_{\mathrm{BT}}(\omega)\big)_+ \leq \eta\Big(\beta_I \, I_{\mathrm{AS}}(\omega) + \beta_C \, C_{\mathrm{BT}}(\omega)\Big) + o(\eta),$$

*where $\beta_I := M_{\mathrm{BT}} K_{\mathrm{bt},I}$, $\beta_C := M_{\mathrm{BT}} K_{\mathrm{bt},C}$.*

We now combine the channel-wise projected drift bounds into a unified two-dimensional description of the local training dynamics within the locking regime.

**Theorem B.15** (2D one-sided self-locking). *Fix $\delta, \varepsilon > 0$ and consider $\omega \in \mathcal{R}_{\delta,\varepsilon}$. Under the conclusions of Propositions B.13 and B.14, the one-sided projected drifts satisfy the following componentwise inequality:*

$$\begin{pmatrix} \Delta_{\mathrm{as}}^+ I_{\mathrm{AS}}(\omega) \\ \Delta_{\mathrm{bt}}^+ C_{\mathrm{BT}}(\omega) \end{pmatrix} \preceq \eta \begin{pmatrix} 0 & \alpha \\ \beta_I & \beta_C \end{pmatrix} \begin{pmatrix} I_{\mathrm{AS}}(\omega) \\ C_{\mathrm{BT}}(\omega) \end{pmatrix} + o(\eta),$$

*where $\preceq$ denotes elementwise inequality.*

**Remark B.16.** *Theorem [B.15](#) controls $\Delta_{\mathrm{as}}^+$ and $\Delta_{\mathrm{bt}}^+$, i.e., the positive parts of the projected drifts. Note that our theoretical result does not assert that $I_{\mathrm{AS}}$ or $C_{\mathrm{BT}}$ cannot decrease, oscillate, or be affected by algorithmic stabilizers (entropy bonuses, clipping, weight decay, etc.). Rather, it formalizes self-locking as the **absence of a strong upward training signal** inside the low-information and low-BT regime: even when improvements happen, the theorem shows they can only be of small magnitude, scaling linearly with the current levels $I_{\mathrm{AS}}$ and $C_{\mathrm{BT}}$.*

**Remark B.17.** *The AS-side drift bound is purely BT-limited: $\Delta_{\mathrm{as}}^+ I_{\mathrm{AS}}(\omega) \lesssim \eta\, C_{\mathrm{BT}}(\omega)$. Thus, when BT is weak, the action-selection kernel receives only a weak positive signal to increase informativeness (consistent with the intuition that AS learning is masked by BT failures in Sec. [2.3](#)). In contrast, the BT-side drift bound is **non-dual**: $\Delta_{\mathrm{bt}}^+ C_{\mathrm{BT}}(\omega) \lesssim \eta\big(I_{\mathrm{AS}}(\omega) + C_{\mathrm{BT}}(\omega)\big)$, where the additional self-term $C_{\mathrm{BT}}(\omega)$ captures a "self-improvement" channel: even under limited information supply, the BT mechanism can still realize some gains by better utilizing the same evidence. At the same time, the presence of $\beta_I\, I_{\mathrm{AS}}(\omega)$ formalizes the bottleneck: when AS information supply stays low, BT improvements cannot scale beyond the evidence-limited envelope, yielding the empirically observed pattern that BT may improve early but tends to plateau once AS remains uninformative, as mentioned in Sec. [2.3](#).*

**Proposition B.18.** *Fix $\delta, \varepsilon > 0$ and let $\omega_0 \in \mathcal{R}_{\delta,\varepsilon}$. Let $m := \max\{\alpha,\ \beta_I + \beta_C\}$. Define*

$$
K := \left\lfloor \frac{1}{\eta m} \log\left( \frac{\varepsilon + \rho\eta}{I_{\mathrm{AS}}(\omega_0) + \rho\eta} \right) \right\rfloor_+ .
$$

*Then any projected evolution consistent with the one-step drift bounds of Theorem [B.15](#) cannot leave $\mathcal{R}_{\delta,\varepsilon}$ within the first $K$ steps.*

## B.5. Why AREW breaks Information Self-locking

In this part, we characterize how and when AREW proposed in Sec. [4](#) breaks information self-locking effectively. For simplicity, we specialize to the AS side and consider only binary critiques $z_t \in \{+1, -1\}$ on action-selection steps. The analysis for the BT side is entirely analogous and omitted for brevity.

**Critique quality and oracle-good actions.** Recall that AS informativeness is measured by the index $I_{\mathrm{AS}}(\omega)$, defined as the expected total improvement in oracle belief quality under the oracle-belief process (Definition [B.2](#)). As discussed before, an action can be assessed independently of the agent's belief-tracking mechanism $\pi_\omega^{\mathrm{bt}}$ by its effect on the terminal oracle confidence $\Psi(\bar{b}_H)$ under the oracle-belief dynamics. Specifically, we give the following definition:

**Definition B.19** (Oracle-good action). *Under the setting of Appendix [B.1](#), consider an oracle-belief trajectory $\bar{\tau} = (\bar{b}_0, \bar{a}_0, \bar{o}_0, \ldots, \bar{b}_H)$ induced by the action-selection kernel $\pi_\omega^{\mathrm{as}}$ and Bayesian belief dynamics $\mathsf{BayesUpd}(\cdot)$. Define the expected terminal oracle confidence following an action $a$ at oracle belief $b$ as $m_t(b, a) := \mathbb{E}\big[\Psi(\bar{b}_H) \mid \bar{b}_t = b,\ \bar{a}_t = a\big]$. An action is oracle-good at step $t$ if it yields a higher terminal oracle confidence than the policy average at the same oracle belief, i.e.,*

$$
m_t(\bar{b}_t, \bar{a}_t) > \mathbb{E}_{a' \sim \pi_\omega^{\mathrm{as}}(\cdot | \bar{b}_t)}\big[m_t(\bar{b}_t, a')\big] .
$$

By telescoping of oracle belief progress, $\sum_{k=t}^{H-1} \Delta\bar{\Psi}_k = \Psi(\bar{b}_H) - \Psi(\bar{b}_t)$, the stepwise *AS advantage with respect to $I_{\mathrm{AS}}$* is given by

$$
\bar{A}_t(\bar{b}_t, \bar{a}_t) = m_t(\bar{b}_t, \bar{a}_t) - \mathbb{E}_{a' \sim \pi_\omega^{\mathrm{as}}(\cdot | \bar{b}_t)}\big[m_t(\bar{b}_t, a')\big] ,
$$

which measures how much the chosen action improves the expected terminal oracle confidence relative to the policy's average choice at the same oracle belief. We therefore define the *oracle direction label*

$$
y_t := \mathrm{sign}\big(\bar{A}_t(\bar{b}_t, \bar{a}_t)\big) \in \{+1, -1\},
$$

indicating whether the step-$t$ action contributes positively or negatively to $I_{\mathrm{AS}}$ under the oracle-belief dynamics.

**Efficacy of AREW and weighted accuracy.** The critique label $z_t \in \{+1, -1\}$ can be viewed as an approximation of the oracle direction label $y_t$. For critique injection to improve AS informativeness beyond outcome-only learning, it is necessary that the injected signal aligns, on average, with the true stepwise contributions to $I_{\mathrm{AS}}$. In particular, positive critique should be assigned more frequently to oracle-good actions than to oracle-bad ones. Moreover, different steps along the trajectory do not contribute equally to AS informativeness. This leads naturally to a notion of **weighted accuracy** of the critique labels with respect to the oracle direction labels. The following proposition shows that, under a mild regularity condition, the effectiveness of AREW in improving AS informativeness is determined by this weighted accuracy.

**Proposition B.20** (Weighted accuracy characterizes AREW improvement). *Under the setting of Sec. 4.2 and Def. B.19, denote the step weight as $w_t(\omega) := |u_t| \left| \bar{A}_t(\bar{b}_t, \bar{a}_t) \right| \| \nabla_\omega \log \pi_\omega^{\mathrm{as}}(\bar{a}_t \mid \bar{b}_t) \|^2$ and let $W(\omega) := \mathbb{E} \left[ \sum_{t=0}^{H-1} w_t(\omega) \right]$. Then the critique quality is measured by the* weighted accuracy

$$\mathrm{Acc}_{\mathrm{as}}(\omega) := \frac{\mathbb{E}\left[ \sum_{t=0}^{H-1} w_t(\omega) \mathbf{1}\{z_t = y_t\} \right]}{\mathbb{E}\left[ \sum_{t=0}^{H-1} w_t(\omega) \right]} \in [0, 1]. \tag{4}$$

*Moreover, the first-order improvement in AS informativeness induced by* AREW *satisfies*

$$I_{\mathrm{AS}}\big(\widehat{\mathcal{T}}_{\mathrm{as}}(\omega)\big) - I_{\mathrm{AS}}\big(\mathcal{T}_{\mathrm{as}}(\omega)\big) = \eta \, W(\omega) \left( 2\,\mathrm{Acc}_{\mathrm{as}}(\omega) - 1 \right) + o(\eta). \tag{5}$$

In particular, when $W(\omega) > 0$, AREW improves AS informativeness beyond the baseline update if and only if $\mathrm{Acc}_{\mathrm{as}}(\omega) > \frac{1}{2}$.

## B.6. Proofs

### B.6.1. PROOF OF PROPOSITION B.11

*Proof.* Fix a turn $t \in \{0, \ldots, H-1\}$ and condition on the current on-policy model belief $b_t = b$. For any action $a \in \mathcal{A}$, define the one-step next belief $b_{t+1}^{(a)}$ as the belief obtained by executing $a_t = a$, observing $o_t = \mathsf{O}(s^\star, a)$ (deterministic), and sampling the BT update $b_{t+1}^{(a)} \sim \pi_\omega^{\mathrm{bt}}(\cdot \mid c_{t+1})$ with $c_{t+1} = (b, a, o_t)$. Define the corresponding one-step potential change $\Delta\Psi_t^{(a)} := \Psi(b_{t+1}^{(a)}) - \Psi(b)$, and its one-sided parts $(\Delta\Psi_t^{(a)})_+$ and $(-\Delta\Psi_t^{(a)})_+$. By the scalar identity $x = (x)_+ - (-x)_+$ applied to $x = \Delta\Psi_t^{(a)}$, we have the decomposition

$$\Delta\Psi_t^{(a)} = (\Delta\Psi_t^{(a)})_+ - (-\Delta\Psi_t^{(a)})_+,$$

equivalently,

$$\Psi(b_{t+1}^{(a)}) = \Psi(b) + (\Delta\Psi_t^{(a)})_+ - (-\Delta\Psi_t^{(a)})_+.$$

Taking conditional expectations given $b_t = b$ and $a_t = a$ yields

$$\mathbb{E}_\omega\left[ \Psi(b_{t+1}^{(a)}) \mid b_t = b, \, a_t = a \right] = \Psi(b) + \mathbb{E}_\omega\left[ (\Delta\Psi_t^{(a)})_+ \mid b_t = b, \, a_t = a \right] - \mathbb{E}_\omega\left[ (-\Delta\Psi_t^{(a)})_+ \mid b_t = b, \, a_t = a \right]. \tag{6}$$

Assumption B.4 states that, for any $a$,

$$\mathbb{E}_\omega\left[ (-\Delta\bar{\Psi}_t^{(a)})_+ \mid b_t = b, \, a_t = a \right] = \mathbb{E}_\omega\left[ \Delta\Psi_t^- \mid b_t = b \right],$$

which is independent of $a$. Therefore, subtracting equation 6 for two actions $a, a'$ gives

$$\mathbb{E}_\omega\left[ \Psi(b_{t+1}^{(a)}) \mid b, a \right] - \mathbb{E}_\omega\left[ \Psi(b_{t+1}^{(a')}) \mid b, a' \right] = \mathbb{E}_\omega\left[ (\Delta\Psi_t^{(a)})_+ \mid b, a \right] - \mathbb{E}_\omega\left[ (\Delta\Psi_t^{(a')})_+ \mid b, a' \right]. \tag{7}$$

By Assumption B.5, along on-policy rollouts,

$$\mathbb{E}_\omega\left[ \Delta\Psi_t^+ \mid b_t, \, a_t \right] \leq \kappa_{\mathrm{bt}} \, C_{\mathrm{BT}}(\omega).$$

Thus, for any realized $b_t = b$ and any $a$ in the support of $\pi_\omega^{\mathrm{as}}(\cdot \mid b)$,

$$\mathbb{E}_\omega\left[ (\Delta\Psi_t^{(a)})_+ \mid b_t = b, \, a_t = a \right] \leq \kappa_{\mathrm{bt}} \, C_{\mathrm{BT}}(\omega). \tag{8}$$

Combining Eq. 7 and 8 yields the bound on the range of the conditional mean next-step confidence:

$$\sup_a \mathbb{E}_\omega\left[ \Psi(b_{t+1}^{(a)}) \mid b_t = b, \, a_t = a \right] - \inf_a \mathbb{E}_\omega\left[ \Psi(b_{t+1}^{(a)}) \mid b_t = b, \, a_t = a \right] \leq \kappa_{\mathrm{bt}} \, C_{\mathrm{BT}}(\omega), \tag{9}$$

where $a \in \mathrm{supp}(\pi_\omega^{\mathrm{as}}(\cdot \mid b))$.

Now define the conditional mean terminal confidence given the current belief at time $t + 1$:

$$G_{t+1,H}(b') := \mathbb{E}_\omega[\Psi(b_H) \mid b_{t+1} = b'].$$

Assumption B.7 implies that $G_{t+1,H}$ is $\kappa_{\mathrm{pr}}$-Lipschitz with respect to $\Psi(\cdot)$: for any realizable $b', b''$,

$$\left|G_{t+1,H}(b') - G_{t+1,H}(b'')\right| \leq \kappa_{\mathrm{pr}} \left|\Psi(b') - \Psi(b'')\right|. \tag{10}$$

For the final-step case $t + 1 = H$, the same bound holds after enlarging $\kappa_{\mathrm{pr}}$ if necessary. Now define the action-conditioned mean terminal confidence at time $t$:

$$m_H(a) := \mathbb{E}_\omega[\Psi(b_H) \mid b_t = b, \ a_t = a].$$

By the tower property,

$$m_H(a) = \mathbb{E}_\omega\left[G_{t+1,H}(b_{t+1}^{(a)}) \ \middle| \ b_t = b, \ a_t = a\right].$$

At this point, we use the fact that $G_{t+1,H}$ is $\kappa_{\mathrm{pr}}$-Lipschitz in $\Psi$ and combine it with the cancellation structure already captured in equation 9: the induced range of $m_H(a)$ over $a$ is bounded by applying equation 10 to the extremal conditional means, giving

$$\sup_a m_H(a) - \inf_a m_H(a) \leq \kappa_{\mathrm{pr}} \left(\sup_a \mathbb{E}_\omega[\Psi(b_{t+1}^{(a)}) \mid b_t = b, \ a_t = a] - \inf_a \mathbb{E}_\omega[\Psi(b_{t+1}^{(a)}) \mid b_t = b, \ a_t = a]\right). \tag{11}$$

Combining equation 11 with equation 9 yields

$$\sup_a m_H(a) - \inf_a m_H(a) \leq \kappa_{\mathrm{pr}} \kappa_{\mathrm{bt}} C_{\mathrm{BT}}(\omega). \tag{12}$$

By Assumption B.1, the outcome value is an $L_R$-Lipschitz function of terminal belief quality, so we write

$$Q_t^{\mathrm{as}}(b, a) = f(m_H(a)).$$

Therefore,

$$\sup_a Q_t^{\mathrm{as}}(b, a) - \inf_a Q_t^{\mathrm{as}}(b, a) \leq L_R \left(\sup_a m_H(a) - \inf_a m_H(a)\right).$$

Plugging equation 12 gives

$$\sup_a Q_t^{\mathrm{as}}(b, a) - \inf_a Q_t^{\mathrm{as}}(b, a) \leq L_R \kappa_{\mathrm{pr}} \kappa_{\mathrm{bt}} C_{\mathrm{BT}}(\omega). \tag{13}$$

By definition, $V_t^{\mathrm{as}}(b) = \mathbb{E}_{a \sim \pi_\omega^{\mathrm{as}}(\cdot \mid b)}[Q_t^{\mathrm{as}}(b, a)]$ is a convex combination of $\{Q_t^{\mathrm{as}}(b, a)\}_a$. Hence for any executed $a_t$,

$$\left|A_t^{\mathrm{as}}(b, a_t)\right| = \left|Q_t^{\mathrm{as}}(b, a_t) - V_t^{\mathrm{as}}(b)\right| \leq \sup_a Q_t^{\mathrm{as}}(b, a) - \inf_a Q_t^{\mathrm{as}}(b, a).$$

Combining with equation 13 yields

$$\left|A_t^{\mathrm{as}}(b_t, a_t)\right| \leq L_R \kappa_{\mathrm{pr}} \kappa_{\mathrm{bt}} C_{\mathrm{BT}}(\omega),$$

and hence the claimed expectation bound.

By definition of the normalized AS-channel update direction,

$$g_{\mathrm{as}}(\omega) = \mathbb{E}_{\tau \sim \Pi_\omega}\left[\frac{1}{H} \sum_{t=0}^{H-1} \nabla_\omega \log \pi_\omega^{\mathrm{as}}(a_t \mid b_t) A_t^{\mathrm{as}}(b_t, a_t)\right].$$

Taking norms and applying Jensen and the triangle inequality,

$$\|g_{\mathrm{as}}(\omega)\| \leq \frac{1}{H} \sum_{t=0}^{H-1} \mathbb{E}\left[\left\|\nabla_\omega \log \pi_\omega^{\mathrm{as}}(a_t \mid b_t)\right\| \cdot \left|A_t^{\mathrm{as}}(b_t, a_t)\right|\right].$$

Using Assumption B.10 gives $\|\nabla_\omega \log \pi_\omega^{\mathrm{as}}(\cdot)\| \leq G_{\mathrm{as}}$ a.s., hence

$$\|g_{\mathrm{as}}(\omega)\| \leq G_{\mathrm{as}} L_R \kappa_{\mathrm{pr}} \kappa_{\mathrm{bt}} C_{\mathrm{BT}}(\omega).$$

This completes the proof with $K_{\mathrm{as}} := G_{\mathrm{as}} L_R \kappa_{\mathrm{pr}} \kappa_{\mathrm{bt}}$. $\qquad\square$

### B.6.2. PROOF OF PROPOSITION B.12

*Proof.* Fix a turn $t$. Condition on the BT context $c_t$ and draw two independent samples $B, B' \overset{\text{i.i.d.}}{\sim} \pi_\omega^{\text{bt}}(\cdot \mid c_t)$. Let

$$Y := Q_t^{\text{bt}}(c_t, B), \qquad Y' := Q_t^{\text{bt}}(c_t, B'), \qquad \bar{Y} := \mathbb{E}[Y \mid c_t] = V_t^{\text{bt}}(c_t).$$

Then $A_t^{\text{bt}}(c_t, B) = Y - \bar{Y}$, and for any fixed realization $Y = y$, Jensen implies $\mathbb{E}[|y - Y'| \mid y, c_t] \geq |y - \mathbb{E}[Y' \mid c_t]| = |y - \bar{Y}|$. Taking expectation over $Y$ yields

$$\mathbb{E}[|Y - \bar{Y}| \mid c_t] \leq \mathbb{E}[|Y - Y'| \mid c_t].$$

Therefore,

$$\mathbb{E}_\omega[|A_t^{\text{bt}}(c_t, b_t)|] \leq \mathbb{E}[|Q_t^{\text{bt}}(c_t, B) - Q_t^{\text{bt}}(c_t, B')|]. \tag{14}$$

By Assumption B.1, $\mathbb{E}[R(\tau) \mid b_H] = f(\Psi(b_H))$ for a non-decreasing $L_R$-Lipschitz $f$. Hence, for any $(c_t, b)$,

$$Q_t^{\text{bt}}(c_t, b) = \mathbb{E}[f(\Psi(b_H)) \mid c_t, \, b_t = b].$$

For any $b, b'$, using $|\mathbb{E}[X] - \mathbb{E}[Y]| \leq \mathbb{E}[|X - Y|]$, Lipschitzness of $f$, and the elementary inequality $|x - y| \leq x + y$ for $x, y \in [0, 1]$, we obtain

$$|Q_t^{\text{bt}}(c_t, b) - Q_t^{\text{bt}}(c_t, b')| \leq L_R\Big(\mathbb{E}[\Psi(b_H) \mid c_t, b_t = b] + \mathbb{E}[\Psi(b_H) \mid c_t, b_t = b']\Big). \tag{15}$$

Along any continuation after time $t$,

$$\Psi(b_H) = \Psi(b_t) + \sum_{k=t}^{H-1} \big(\Psi(b_{k+1}) - \Psi(b_k)\big) \leq \Psi(b_t) + \sum_{k=t}^{H-1} \Delta\Psi_k^+.$$

Taking conditional expectation given $(c_t, b_t = b)$ yields

$$\mathbb{E}[\Psi(b_H) \mid c_t, b_t = b] \leq \Psi(b) + \mathbb{E}\left[\sum_{k=t}^{H-1} \Delta\Psi_k^+ \,\middle|\, c_t, b_t = b\right]. \tag{16}$$

By the evidence-limited interpretation of absorbed belief progress, the future positive model-belief progress from $b$ is bounded by the residual oracle AS information budget from the same belief:

$$\mathbb{E}\left[\sum_{k=t}^{H-1} \Delta\Psi_k^+ \,\middle|\, c_t, b_t = b\right] \leq V_{\text{AS},t}^{\text{orc}}(b). \tag{17}$$

Assumption B.6 further gives

$$V_{\text{AS},t}^{\text{orc}}(b) \leq \kappa_{\text{or}} I_{\text{AS}}(\omega). \tag{18}$$

Combining equation 17 and equation 18, we obtain

$$\mathbb{E}\left[\sum_{k=t}^{H-1} \Delta\Psi_k^+ \,\middle|\, c_t, b_t = b\right] \leq \kappa_{\text{or}} I_{\text{AS}}(\omega). \tag{19}$$

Combining equation 14, equation 15, equation 16, and equation 19, and averaging over $B, B'$, we obtain

$$\mathbb{E}[|A_t^{\text{bt}}(c_t, b_t)|] \leq 2L_R\Big(\mathbb{E}[\Psi(b_t)] + \kappa_{\text{or}} I_{\text{AS}}(\omega)\Big).$$

It remains to control $\mathbb{E}[\Psi(b_t)]$ by $C_{\text{BT}}(\omega)$ up to the fixed initial potential. Along any rollout,

$$\Psi(b_t) = \Psi(b_0) + \sum_{k=0}^{t-1} \big(\Psi(b_{k+1}) - \Psi(b_k)\big) \leq \Psi(b_0) + \sum_{k=0}^{t-1} \Delta\Psi_k^+ \leq \Psi(b_0) + \sum_{k=0}^{H-1} \Delta\Psi_k^+.$$

Taking expectation yields $\mathbb{E}[\Psi(b_t)] \le \mathbb{E}[\Psi(b_0)] + C_{\mathrm{BT}}(\omega)$. Since the initial potential is fixed by the task context and contributes only a common baseline to the BT-channel advantage, we absorb it into the baseline normalization and obtain the stated bound

$$\mathbb{E}_\omega\big[|A_t^{\mathrm{bt}}(c_t, b_t)|\big] \;\le\; 2L_R\Big(C_{\mathrm{BT}}(\omega) + \kappa_{\mathrm{or}}\, I_{\mathrm{AS}}(\omega)\Big).$$

By definition of the normalized BT-channel update direction,

$$g_{\mathrm{bt}}(\omega) = \mathbb{E}_{\tau \sim \Pi_\omega}\left[\frac{1}{H+1}\sum_{t=0}^{H} \nabla_\omega \log \pi_\omega^{\mathrm{bt}}(b_t \mid c_t)\, A_t^{\mathrm{bt}}(c_t, b_t)\right].$$

Taking norms, applying Jensen, and using Assumption B.10 ($\|\nabla_\omega \log \pi_\omega^{\mathrm{bt}}\| \le G_{\mathrm{bt}}$ a.s.) gives

$$\|g_{\mathrm{bt}}(\omega)\| \le \frac{1}{H+1}\sum_{t=0}^{H} \mathbb{E}\big[\|\nabla_\omega \log \pi_\omega^{\mathrm{bt}}(b_t \mid c_t)\| \cdot |A_t^{\mathrm{bt}}|\big] \le 2G_{\mathrm{bt}}L_R\Big(C_{\mathrm{BT}}(\omega) + \kappa_{\mathrm{or}}\, I_{\mathrm{AS}}(\omega)\Big).$$

Therefore

$$\|g_{\mathrm{bt}}(\omega)\| \le K_{\mathrm{bt},I}\, I_{\mathrm{AS}}(\omega) + K_{\mathrm{bt},C}\, C_{\mathrm{BT}}(\omega),$$

with $K_{\mathrm{bt},I} := 2G_{\mathrm{bt}}L_R\kappa_{\mathrm{or}}$ and $K_{\mathrm{bt},C} := 2G_{\mathrm{bt}}L_R$. $\qquad\square$

### B.6.3. PROOF OF PROPOSITION B.13

*Proof.* Fix $\omega \in \mathcal{R}_{\delta,\varepsilon}$. Recall $\mathcal{T}_{\mathrm{as}}(\omega) = \omega + \eta g_{\mathrm{as}}(\omega)$. By Assumption B.9 (first-order expandability of $I_{\mathrm{AS}}$ on $\mathcal{R}_{\delta,\varepsilon}$),

$$I_{\mathrm{AS}}(\mathcal{T}_{\mathrm{as}}(\omega)) - I_{\mathrm{AS}}(\omega) = \Big\langle \nabla_\omega I_{\mathrm{AS}}(\omega),\, \mathcal{T}_{\mathrm{as}}(\omega) - \omega \Big\rangle + o(\eta) = \eta\langle \nabla_\omega I_{\mathrm{AS}}(\omega),\, g_{\mathrm{as}}(\omega)\rangle + o(\eta).$$

Taking the positive part and using $(x+y)_+ \le x_+ + |y|$ yields

$$\Delta_{\mathrm{as}}^+ I_{\mathrm{AS}}(\omega) = \Big(\eta\langle \nabla_\omega I_{\mathrm{AS}}(\omega),\, g_{\mathrm{as}}(\omega)\rangle + o(\eta)\Big)_+ \le \eta\Big(\langle \nabla_\omega I_{\mathrm{AS}}(\omega),\, g_{\mathrm{as}}(\omega)\rangle\Big)_+ + |o(\eta)|.$$

Next, apply Cauchy–Schwarz and the gradient bound $\|\nabla_\omega I_{\mathrm{AS}}(\omega)\| \le M_{\mathrm{AS}}$:

$$\Big(\langle \nabla_\omega I_{\mathrm{AS}}(\omega),\, g_{\mathrm{as}}(\omega)\rangle\Big)_+ \le \big|\langle \nabla_\omega I_{\mathrm{AS}}(\omega),\, g_{\mathrm{as}}(\omega)\rangle\big| \le \|\nabla_\omega I_{\mathrm{AS}}(\omega)\| \cdot \|g_{\mathrm{as}}(\omega)\| \le M_{\mathrm{AS}}\, \|g_{\mathrm{as}}(\omega)\|.$$

Using the norm control $\|g_{\mathrm{as}}(\omega)\| \le K_{\mathrm{as}}C_{\mathrm{BT}}(\omega)$ gives

$$\Delta_{\mathrm{as}}^+ I_{\mathrm{AS}}(\omega) \le \eta\, M_{\mathrm{AS}}\, K_{\mathrm{as}}\, C_{\mathrm{BT}}(\omega) + |o(\eta)|.$$

Finally, absorb $|o(\eta)|$ into $o(\eta)$ (since $|o(\eta)|/\eta \to 0$ uniformly on $\mathcal{R}_{\delta,\varepsilon}$) to obtain the claimed bound with $\alpha := M_{\mathrm{AS}}K_{\mathrm{as}}$. $\qquad\square$

### B.6.4. PROOF OF PROPOSITION B.14

*Proof.* Fix $\omega \in \mathcal{R}_{\delta,\varepsilon}$. Recall $\mathcal{T}_{\mathrm{bt}}(\omega) = \omega + \eta g_{\mathrm{bt}}(\omega)$. By Assumption B.9 (first-order expandability of $C_{\mathrm{BT}}$ on $\mathcal{R}_{\delta,\varepsilon}$),

$$C_{\mathrm{BT}}(\mathcal{T}_{\mathrm{bt}}(\omega)) - C_{\mathrm{BT}}(\omega) = \Big\langle \nabla_\omega C_{\mathrm{BT}}(\omega),\, \mathcal{T}_{\mathrm{bt}}(\omega) - \omega \Big\rangle + o(\eta) = \eta\langle \nabla_\omega C_{\mathrm{BT}}(\omega),\, g_{\mathrm{bt}}(\omega)\rangle + o(\eta).$$

Taking positive parts and using $(x+y)_+ \le x_+ + |y|$ gives

$$\Delta_{\mathrm{bt}}^+ C_{\mathrm{BT}}(\omega) = \Big(\eta\langle \nabla_\omega C_{\mathrm{BT}}(\omega),\, g_{\mathrm{bt}}(\omega)\rangle + o(\eta)\Big)_+ \le \eta\Big(\langle \nabla_\omega C_{\mathrm{BT}}(\omega),\, g_{\mathrm{bt}}(\omega)\rangle\Big)_+ + |o(\eta)|.$$

Apply Cauchy–Schwarz and the gradient bound $\|\nabla_\omega C_{\mathrm{BT}}(\omega)\| \le M_{\mathrm{BT}}$:

$$\Big(\langle \nabla_\omega C_{\mathrm{BT}}(\omega),\, g_{\mathrm{bt}}(\omega)\rangle\Big)_+ \le \big|\langle \nabla_\omega C_{\mathrm{BT}}(\omega),\, g_{\mathrm{bt}}(\omega)\rangle\big| \le \|\nabla_\omega C_{\mathrm{BT}}(\omega)\| \cdot \|g_{\mathrm{bt}}(\omega)\| \le M_{\mathrm{BT}}\, \|g_{\mathrm{bt}}(\omega)\|.$$

Using the norm control on $\mathcal{R}_{\delta,\varepsilon}$,

$$\|g_{\mathrm{bt}}(\omega)\| \;\le\; K_{\mathrm{bt},I}\, I_{\mathrm{AS}}(\omega) \;+\; K_{\mathrm{bt},C}\, C_{\mathrm{BT}}(\omega),$$

we obtain

$$\Delta_{\mathrm{bt}}^+ C_{\mathrm{BT}}(\omega) \le \eta\, M_{\mathrm{BT}}\Big(K_{\mathrm{bt},I}\, I_{\mathrm{AS}}(\omega) + K_{\mathrm{bt},C}\, C_{\mathrm{BT}}(\omega)\Big) + |o(\eta)|.$$

Absorbing $|o(\eta)|$ into $o(\eta)$ yields the stated bound with $\beta_I := M_{\mathrm{BT}}K_{\mathrm{bt},I}$ and $\beta_C := M_{\mathrm{BT}}K_{\mathrm{bt},C}$. $\qquad\square$

### B.6.5. PROOF OF PROPOSITION B.18

*Proof.* Since the $o(\eta)$ term in Theorem B.15 is uniform over $\mathcal{R}_{\delta,\varepsilon}$, there exist $\eta_0 > 0$ and $c_0 < \infty$ such that for all $\eta \in (0, \eta_0]$ and all $\omega \in \mathcal{R}_{\delta,\varepsilon}$, the remainder satisfies the componentwise bound $\|o(\eta)\|_\infty \leq c_0 \eta^2$. Set $\rho := c_0/m$.

Define the capability vector

$$\mathbf{x}(\omega) := \begin{pmatrix} I_{\mathrm{AS}}(\omega) \\ C_{\mathrm{BT}}(\omega) \end{pmatrix} \in \mathbb{R}_+^2, \quad M := \begin{pmatrix} 0 & \alpha \\ \beta_I & \beta_C \end{pmatrix}, \quad \mathbf{1} := (1,1)^\top.$$

Inside $\mathcal{R}_{\delta,\varepsilon}$, Theorem B.15 yields the componentwise bound

$$\Delta^+ \mathbf{x}(\omega) \preceq \eta M \mathbf{x}(\omega) + c_0 \eta^2 \mathbf{1}.$$

Moreover, for any scalar $z$ we have $z \leq z_+$, hence any increment in $\mathbf{x}$ is bounded above by its positive part. Therefore, an upper envelope for the accumulation of positive gains is given by the deterministic recursion

$$\mathbf{x}_{k+1} \preceq (I + \eta M) \mathbf{x}_k + c_0 \eta^2 \mathbf{1}, \qquad \mathbf{x}_0 := \mathbf{x}(\omega_0). \tag{20}$$

Let $y_k := \|\mathbf{x}_k\|_\infty$. Taking $\|\cdot\|_\infty$ on equation 20 and using $\|I + \eta M\|_\infty = 1 + \eta m$ gives

$$y_{k+1} \leq (1 + \eta m) y_k + c_0 \eta^2.$$

Unrolling this scalar recursion and using $(1 + \eta m)^k \leq e^{k\eta m}$ yields

$$y_k \leq e^{k\eta m}\left(y_0 + \tfrac{c_0}{m}\eta\right) - \tfrac{c_0}{m}\eta = e^{k\eta m}\left(y_0 + \rho\eta\right) - \rho\eta.$$

Under the stated initialization envelope used in the trapping bound, $y_0 \leq I_{\mathrm{AS}}(\omega_0)$. Hence, if

$$e^{k\eta m}\left(I_{\mathrm{AS}}(\omega_0) + \rho\eta\right) - \rho\eta \leq \varepsilon,$$

then $y_k \leq \varepsilon$, which implies $\mathbf{x}_k \preceq (\varepsilon, \varepsilon)^\top \preceq (\delta, \varepsilon)^\top$ whenever $\delta \geq \varepsilon$. Solving the inequality for $k$ gives

$$k \leq \frac{1}{\eta m} \log\left(\frac{\varepsilon + \rho\eta}{I_{\mathrm{AS}}(\omega_0) + \rho\eta}\right).$$

Choosing

$$K = \left\lfloor \frac{1}{\eta m} \log\left(\frac{\varepsilon + \rho\eta}{I_{\mathrm{AS}}(\omega_0) + \rho\eta}\right) \right\rfloor_+$$

ensures the above condition holds for all integers $k \in \{0, 1, \ldots, K\}$, proving the claimed finite-horizon trapping under the projected drift envelope. $\qquad\square$

### B.7. Proof of Proposition B.20

*Proof.* We absorb the fixed critique strength into the auxiliary direction and write the critique-shaped update as

$$\mathcal{T}_{\mathrm{as}}(\omega) - \omega = \eta\, g_{\mathrm{as}}(\omega), \tag{21}$$

$$\widehat{\mathcal{T}}_{\mathrm{as}}(\omega) - \omega = \eta\, g_{\mathrm{as}}(\omega) + \eta\, g_{\mathrm{aux,as}}(\omega). \tag{22}$$

By Assumption B.9, for $\omega' = \mathcal{T}_{\mathrm{as}}(\omega) \in \mathcal{R}_{\delta,\varepsilon}$ we have

$$I_{\mathrm{AS}}(\mathcal{T}_{\mathrm{as}}(\omega)) = I_{\mathrm{AS}}(\omega) + \left\langle \nabla_\omega I_{\mathrm{AS}}(\omega),\ \mathcal{T}_{\mathrm{as}}(\omega) - \omega \right\rangle + o(\eta), \tag{23}$$

and for $\omega' = \widehat{\mathcal{T}}_{\mathrm{as}}(\omega) \in \mathcal{R}_{\delta,\varepsilon}$ we have

$$I_{\mathrm{AS}}(\widehat{\mathcal{T}}_{\mathrm{as}}(\omega)) = I_{\mathrm{AS}}(\omega) + \left\langle \nabla_\omega I_{\mathrm{AS}}(\omega),\ \widehat{\mathcal{T}}_{\mathrm{as}}(\omega) - \omega \right\rangle + o(\eta). \tag{24}$$

Subtracting equation 23 from equation 24 yields

$$I_{\mathrm{AS}}(\widehat{\mathcal{T}}_{\mathrm{as}}(\omega)) - I_{\mathrm{AS}}(\mathcal{T}_{\mathrm{as}}(\omega)) = \left\langle \nabla_\omega I_{\mathrm{AS}}(\omega),\ \widehat{\mathcal{T}}_{\mathrm{as}}(\omega) - \mathcal{T}_{\mathrm{as}}(\omega) \right\rangle + o(\eta).$$

Using the update definitions, $\widehat{\mathcal{T}}_{\mathrm{as}}(\omega) - \mathcal{T}_{\mathrm{as}}(\omega) = \eta\, g_{\mathrm{aux,as}}(\omega)$, so

$$I_{\mathrm{AS}}(\widehat{\mathcal{T}}_{\mathrm{as}}(\omega)) - I_{\mathrm{AS}}(\mathcal{T}_{\mathrm{as}}(\omega)) = \eta \left\langle \nabla_\omega I_{\mathrm{AS}}(\omega),\ g_{\mathrm{aux,as}}(\omega) \right\rangle + o(\eta)$$
$$=: \eta\, \Gamma_{\mathrm{as}}(\omega)\ + o(\eta). \tag{25}$$

We expand $\Gamma_{\mathrm{as}}(\omega)$. On the oracle-belief process, $I_{\mathrm{AS}}(\omega) = \mathbb{E}_{\bar{\tau}}[\sum_{t=0}^{H-1} \Delta \bar{\Psi}_t]$ is a finite-horizon policy objective with action $\bar{a}_t \sim \pi_\omega^{\mathrm{as}}(\cdot \mid \bar{b}_t)$. Thus, the policy-gradient theorem yields

$$\nabla_\omega I_{\mathrm{AS}}(\omega) = \mathbb{E}_{\bar{\tau}}\left[ \sum_{t=0}^{H-1} \bar{A}_t(\bar{b}_t, \bar{a}_t)\, s_t(\omega) \right], \qquad s_t(\omega) = \nabla_\omega \log \pi_\omega^{\mathrm{as}}(\bar{a}_t \mid \bar{b}_t).$$

The likelihood-gap auxiliary objective in Section 4.2 induces an additive update component of the form

$$g_{\mathrm{aux,as}}(\omega) = \mathbb{E}_{\bar{\tau}}\left[ \sum_{t=0}^{H-1} u_t(z)\, s_t(\omega) \right], \qquad u_t(z) = |u_t|\, z_t, \ \ |u_t| \geq 0, \ \ z_t \in \{+1, -1\}.$$

By bilinearity and exchanging expectation with finite sums,

$$\Gamma_{\mathrm{as}}(\omega)\ \propto\ \mathbb{E}\left[ \left\langle \sum_t \bar{A}_t\, s_t,\ \sum_u u_u(z)\, s_u \right\rangle \right] = \mathbb{E}\left[ \sum_{t,u} \bar{A}_t\, u_u(z)\, \langle s_t, s_u \rangle \right].$$

Under the diagonal score-covariance approximation used for this first-order characterization, we keep the same-time terms, giving

$$\Gamma_{\mathrm{as}}(\omega) = \mathbb{E}\left[ \sum_{t=0}^{H-1} \bar{A}_t(\bar{b}_t, \bar{a}_t)\, u_t(z)\, \|s_t(\omega)\|^2 \right].$$

Write $\bar{A}_t = |\bar{A}_t|\, y_t$ with $y_t = \mathrm{sign}(\bar{A}_t) \in \{\pm 1\}$ and $u_t = |u_t|\, z_t$ with $z_t \in \{\pm 1\}$. Then each summand becomes

$$\bar{A}_t\, u_t\, \|s_t\|^2 = |u_t|\, |\bar{A}_t|\, \|s_t\|^2\, (z_t y_t) = w_t(\omega)\, (z_t y_t),$$

where $w_t(\omega)$ is exactly the weight defined above. Hence

$$\Gamma_{\mathrm{as}}(\omega) = \mathbb{E}\left[ \sum_{t=0}^{H-1} w_t(\omega)\, z_t y_t \right].$$

Since $z_t y_t = +1$ iff $z_t = y_t$ and $z_t y_t = -1$ iff $z_t \neq y_t$, we have $z_t y_t = 2\,\mathbf{1}\{z_t = y_t\} - 1$, and therefore

$$\Gamma_{\mathrm{as}}(\omega) = \mathbb{E}\left[ \sum_t w_t(\omega)\, \big(2\,\mathbf{1}\{z_t = y_t\} - 1\big) \right]$$
$$= 2\, \mathbb{E}\left[ \sum_t w_t(\omega)\, \mathbf{1}\{z_t = y_t\} \right] - \mathbb{E}\left[ \sum_t w_t(\omega) \right]$$
$$= W(\omega)\, \big(2\,\mathrm{Acc}_{\mathrm{as}}(\omega) - 1\big),$$

which is equation 5. The final equivalence follows immediately when $W(\omega) > 0$. $\qquad\square$

## C. Setup Details

### C.1. Dataset Details and Prompt Templates

In this section, we present more details for the datasets and tasks evaluated in this work.

**Preference Estimation – Gated (PE-G)**, adapted from Badola et al. (2025) and Zou et al. (2026). Adapted from Badola et al. (2025), Gated-PE is an interactive preference inference task under constrained information acquisition. The agent is given a finite set of items $\mathcal{X} = \{x_1, \ldots, x_N\}$, where each item $x_i$ is represented by a known attribute vector $\mathbf{a}_i \in \mathbb{R}^D$. The user is characterized by an unknown latent preference vector $\mathbf{w}^\star \in [0,1]^D$. Through interaction, the agent maintains and iteratively refines an estimate $\mathbf{w}_t \in [0,1]^D$ of this latent preference. At each decision point, the agent actively selects a low-dimensional attribute subspace $S_t \subseteq \{1, \ldots, D\}$ and an item comparison $(x_i, x_j) \in \mathcal{X} \times \mathcal{X}$ designed to elicit the user's preference feedback restricted to $S_t$. Based on the observed feedback, the agent updates its belief state. The objective is to accurately recover $\mathbf{w}^\star$ under sparse, outcome-based supervision.

**MediQ**, adapted from Li et al. (2024). Adapted from Li et al. (2024), MediQ is an interactive medical inference task that models hypothesis-level belief tracking under partial observability. The agent is provided with a clinical vignette and an associated medical question whose answer lies in a finite hypothesis set of size $D$. The agent maintains a belief state $\mathbf{w}_t \in [0,1]^D$, where each dimension represents the current support for a candidate hypothesis. Through iterative interaction, the agent actively queries the LLM-simulated user for diagnostic information, receives structured feedback, and updates each hypothesis score accordingly. The learning objective is to progressively concentrate belief mass onto the correct hypothesis.

**FloDial**, adapted from Raghu et al. (2021); Hu et al. (2024) consists of multi-turn diagnostic dialogues for resolving user-reported issues. It provides a scenario where a customer support technician interacts with customers to identify and resolve faults or issues within computer systems, electronic devices, machinery, or other complex systems. The agent simulates the customer support technician, which chat with the customer to further check the specific issues of device through multi-turn interactions.

### C.2. Baseline Details

Here we introduce RL algorithms used in our experiments. Formally, given an actor model $\pi_\theta$, the likelihood of a response $y$ to a query $x$ under the policy $\pi_\theta$ is modeled as $\pi_\theta(y|x) = \prod_{t=1}^{|y|} \pi_\theta(y_t|x, y_{<t})$. Given a query-response pair $(x, y)$, a verifier $r$ generates its reward $r(x, y) \in [0, 1]$.

**Proximal Policy Optimization (PPO)** (Schulman et al., 2017) employs the following objective for policy optimization:

$$\mathcal{J}_{\text{PPO}}(\theta) = \mathbb{E}_{x \sim \mathcal{D}, y \sim \pi_{\theta_{\text{old}}}(\cdot|x)} \left[ \frac{1}{|y|} \sum_{t=1}^{|y|} \min \left( w_t(\theta) \widehat{A}_t, \text{clip}\left( w_t(\theta), 1 - \varepsilon, 1 + \varepsilon \right) \widehat{A}_t \right) \right], \tag{26}$$

where the importance ratio of the token $y_t$ is defined as $w_t(\theta) = \frac{\pi_\theta(y_t|x, y_{<t})}{\pi_{\theta_{\text{old}}}(y_t|x, y_{<t})}$, the advantage $\widehat{A}_t$ of $y_t$ is typically computed via Generalized Advantage Estimation (GAE) (Schulman et al., 2015) with temporal-difference errors, and $\varepsilon$ is the clipping range of importance ratios.

**Group Relative Policy Optimization (GRPO)** (Shao et al., 2024) proposes computing the relative advantage of each response within a group of responses of the same query using the following objective (omitting the KL regularization term):

$$\mathcal{J}_{\text{GRPO}}(\theta) = \mathbb{E}_{x, \{y_i\}_{i=1}^G} \left[ \frac{1}{G} \sum_{i=1}^G \frac{1}{|y_i|} \sum_{t=1}^{|y_i|} \min \left( w_{i,t}(\theta) \widehat{A}_{i,t}, \text{clip}\left( w_{i,t}(\theta), 1 - \varepsilon, 1 + \varepsilon \right) \widehat{A}_{i,t} \right) \right], \tag{27}$$

where $\{y_i\}_{i=1}^G \sim \pi_{\theta_{\text{old}}}(\cdot|x)$ and $G$ is the group size. The importance ratio $w_{i,t}(\theta)$ and advantage $\widehat{A}_{i,t}$ of token $y_{i,t}$ are defined as:

$$w_{i,t}(\theta) = \frac{\pi_\theta(y_{i,t}|x, y_{i,<t})}{\pi_{\theta_{\text{old}}}(y_{i,t}|x, y_{i,<t})}, \quad \widehat{A}_{i,t} = \frac{r(x, y_i) - \text{mean}\left( \{r(x, y_i)\}_{i=1}^G \right)}{\text{std}\left( \{r(x, y_i)\}_{i=1}^G \right)}, \tag{28}$$

respectively, where all the tokens in $y_i$ share the same advantage.

**Group Sequence Policy Optimization (GSPO)** (Zheng et al., 2025) extends GRPO by defining the importance ratio at the sequence level with length normalization, with sequence-level clipping, rewarding, and optimization. The objective is:

$$\mathcal{J}_{\text{GSPO}}(\theta) = \mathbb{E}_{x, \{y_i\}_{i=1}^G} \left[ \frac{1}{G} \sum_{i=1}^G \min\left( s_i(\theta)\widehat{A}_i, \ \text{clip}(s_i(\theta), 1-\epsilon, 1+\epsilon)\widehat{A}_i \right) \right], \tag{29}$$

where

$$s_i(\theta) = \left( \frac{\pi_\theta(y_i|x)}{\pi_{\theta_{\text{old}}}(y_i|x)} \right)^{1/|y_i|} = \exp\left( \frac{1}{|y_i|} \sum_{t=1}^{|y_i|} \log \frac{\pi_\theta(y_{i,t}|x, y_{i,<t})}{\pi_{\theta_{\text{old}}}(y_{i,t}|x, y_{i,<t})} \right).$$

### C.3. Supplementary Implementation Details

#### C.3.1. ADDITIONAL SETTING ON DATASETS

Here we provide additional implementation details. The maximum number of interaction turns is set at 10 for PE-G, 12 for PE-F, 8 for MediQ, 10 for FloDial-Easy and FloDial-Hard. For RL training, we define task-specific rewards aligned with their evaluation metrics: for PE-G and PE-F, the reward is the similarity improvement compared to the initial default guess (all 0.5), where PE-G leverages binary reward ($\mathbf{1}[\text{improvement}] > 0.03$) and PE-F leverages the continuous improvement value (normalized to $[0, 1]$). For MediQ and FloDial, the reward is binary, checking if the final decision made by the agents aligns with the ground-truth. All rewards are provided only at the terminal step of each trajectory, consistent with the outcome-based RL setting.

The AS and BT proxies on PE and MediQ datasets can be seen in Sec. 2. For the FloDial dataset, the query is considered as uninformative if the user replies "unknown", and the BT proxy is defined by whether the agent can increase the confidence of the ground truth when receiving informative feedback.

**Critique construction (AS vs. BT).** Across datasets, we use two lightweight stepwise critiques: $z_t^{\text{AS}}$ (AS-channel) evaluates whether the *query step* at turn $t$ yields informative feedback, and $z_t^{\text{BT}}$ (BT-channel) evaluates whether the subsequent *belief update* is consistent with that feedback. Below we summarize the dataset-specific instantiations.

**PE-G and PE-F.**

*AS critique.* Let the action be a comparison between two items, inducing a signed attribute-difference vector $m_{ij} \in \mathbb{R}^D$ (restricted to the focused coordinates when applicable). We mark the query as informative if it exhibits a non-trivial trade-off across coordinates, i.e.,

$$z_t^{\text{AS}} = +1 \iff \exists k, \ell \in [D] \text{ s.t. } m_{ij,k}\, m_{ij,\ell} < 0,$$

and set $z_t^{\text{AS}} = -1$ for repeated or invalid queries or when $m_{ij}$ is one-signed.

*BT critique.* Conditioned on an informative query ($z_t^{\text{AS}} = +1$), we compare the updated preference estimate $v_t$ against the previous valid estimate $v_{t-1}$ using similarity to the ground-truth preference $w^\star$:

$$z_t^{\text{BT}} = \text{sign}\Big( \text{sim}(v_t, w^\star) - \text{sim}(v_{t-1}, w^\star) \Big),$$

and set $z_t^{\text{BT}} = 0$ when the query is uninformative or invalid.

**MediQ.**

*AS critique.* Let $f_t$ denote the patient feedback to the query at turn $t$. We set $z_t^{\text{AS}} = +1$ iff the feedback is informative (i.e., not an "Unknown / cannot answer" response) and the query is not repeated; otherwise $z_t^{\text{AS}} = -1$. We additionally explore counterfactual pattern (exploration on other datasets as future work). When counterfactual queries are enabled, we intervene the belief from the last turn and obtain the counterfactual query. We then discourage semantic repetition by requiring the counterfactual query to be sufficiently different from the actual query (measured by token-level overlap), and mark the step as uninformative when this difference is small.

*BT critique.* Let $w_t \in [0, 1]^4$ be the belief over 4 hypotheses and $s^\star$ the ground-truth hypothesis. Define the margin

$$\text{mar}(w_t; s^\star) := w_t(s^\star) - \max_{s \neq s^\star} w_t(s).$$

If the feedback is uninformative, we enforce *invariance*: $z_t^{\mathrm{BT}} = +1$ if $\mathrm{mar}(w_t; s^\star) = \mathrm{mar}(w_{t-1}; s^\star)$ and $z_t^{\mathrm{BT}} = -1$ otherwise. If the feedback is informative, we employ a counterfactual-consistency check: We intervene the observation to "unknown" and obtain $w_t^{\mathrm{cf}}$ to be a counterfactual update from the same previous belief $w_{t-1}$ but from the "unknown" response; then

$$z_t^{\mathrm{BT}} = +1 \quad \Longleftrightarrow \quad \|w_t^{\mathrm{cf}} - w_{t-1}\|_2 \leq \|w_t - w_{t-1}\|_2,$$

and $z_t^{\mathrm{BT}} = -1$ otherwise.

**FloDial.** *AS critique.* User feedback takes the form `Yes`/`No` when the query matches a reference diagnostic item, and `Unknown` otherwise. We set

$$z_t^{\mathrm{AS}} = \begin{cases} +1, & \text{if feedback is } \texttt{Yes}, \\ 0, & \text{if feedback is } \texttt{No}, \\ -1, & \text{if feedback is } \texttt{Unknown} \text{ (no match)}. \end{cases}$$

*BT critique.* Let $w_t \in [0,1]^S$ be the belief over candidate faults and $s^\star$ the ground-truth fault. When the feedback is informative (`Yes`/`No`), we encourage increasing confidence on $s^\star$:

$$z_t^{\mathrm{BT}} = +1 \iff w_t(s^\star) > w_{t-1}(s^\star), \qquad z_t^{\mathrm{BT}} = -1 \text{ otherwise.}$$

When the feedback is uninformative (`Unknown`), we use an invariance-style rule: we require the belief to remain unchanged (e.g., $z_t^{\mathrm{BT}} = +1$ iff $w_t = w_{t-1}$).

### C.3.2. TRAINING CONFIGURATIONS

All expriments are trained on a single node with 8 B200 GPUs, based on the implementations of Verl (Sheng et al., 2025). All training tasks on PPO are conducted for 200 steps (GRPO and GSPO 100 steps) with the actor model optimized using a learning rate of $1.0 \times 10^{-6}$. For distributed training, we adopt Fully Sharded Data Parallelism (FSDP), using BFloat16 precision throughout both training and evaluation. For efficient LLM rollouts, we adopt vLLM [1] with a tensor parallel size of 1. The rollout sampling uses a temperature of 1.0 for all datasets.

For the PPO baseline, we use Generalized Advantage Estimation (GAE) with parameters $\lambda = 1$ and $\gamma = 1$. The clip ratio $\varepsilon$ are set to 0.2. For GRPO training, we sample 3 responses per prompt, and the rollout parameters with the clip ratio are consistent with the PPO setting. For the GSPO algorithm, the clip ratio $\varepsilon_{low}$ and $\varepsilon_{high}$ are set to 0.0003 and 0.0004, respectively, while others keep consistent with GRPO training.

---

[1] https://docs.vllm.ai/en/latest/

---

### Input Prompts for the MovieRec Preference Estimation Dataset

You are an interactive preference estimation agent. The goal is to infer a user's {len_attributes}-dimensional hidden preference vector on movies through multi-round interaction.

## Setup:
- You are given {len_seen} movies, each with scores on {len_attributes} dimensions (indexed 1 . . . {len_attributes}):
{seen_movie_sample}

- Maintain an estimate of the user's preference vector:
   Guess: $w_1, w_2, \ldots, w_{\{len\_attributes\}}$
- Initialization:
   Guess: $0.5, 0.5, \ldots, 0.5$

## Interaction Protocol:
Interaction alternates between two types of rounds.

1) **Action Round (odd-numbered rounds: 1,3,5,. . . )**
In an Action Round, you must choose which information to query. You must output exactly:

```
<interact>
Focus: k1,k2,k3
Pair: p1,p2
</interact>
```

Rules:
- Focus must contain exactly three distinct integers in $[1, \{len\_attributes\}]$. Ensure that all dimensions receive opportunities to be disambiguated over the course of interaction.
- Pair must contain exactly two distinct integers in $[1, \{len\_seen\}]$.
- Avoid uninformative pairs, such as dominance pairs where one movie is better on all focused dimensions $k_1, k_2, k_3$.

After you output Focus and Pair, the user will provide feedback:

- User Feedback: "Yes", "No", or "Equal".
   **"Yes" means $p_1$ is preferred when considering only dimensions $k_1, k_2, k_3$.**

2) **Update Round (even-numbered rounds: 2,4,6,. . . )**
In an Update Round, you must update the preference estimate based on the most recent Focus, Pair, and User Feedback.
You must output exactly:

```
<interact>
Guess: w1,w2,...,w{len_attributes}
</interact>
```

Rules:
- Guess must be comma-separated numbers.
- Use the feedback to adjust the relative importance of the focused dimensions in a way consistent with the observed preference.

## Reasoning:
Before each `<interact>` block, you may briefly reason about what to do in a:

```
<scratch>... </scratch>
```

Round 1 is an Action Round.

*Figure 8.* Prompt Template for MovieRec Preference Estimation.

---

### Input Prompts for the MediQ Dataset

You are an interactive medical inference agent. Your goal is to iteratively maintain and refine a 4-dimensional state vector that tracks the relative support for four potential hypotheses associated with a given medical question, through multi-round interaction.

## Setup:
- You are given: (i) a clinical vignette describing a patient scenario, and (ii) a medical question associated with this scenario.
- The question admits four potential hypotheses, labeled A, B, C, and D.
- Maintain a state vector $(w_A, w_B, w_C, w_D)$, where each $w \in [0, 1]$, representing your current level of support for each hypothesis.
- Initialization:
    Guess: $0.5, 0.5, 0.5, 0.5$
- In each Action round, you can issue **only one** query. The query must be a single, atomic question intended to reduce uncertainty among the four hypotheses; compound or multi-part questions are not allowed.

## Alternating Interaction Protocol:
Interaction alternates between two types of rounds:

1) **Action Round (odd-numbered rounds: 1,3,5,...)**
- You must output only a query in the exact format:

```
<interact>
Query: ...
</interact>
```

- After you output a Query, the user will provide feedback.

2) **Update Round (even-numbered rounds: 2,4,6,...)**
- You must update and output only the state vector (Guess) based on: (a) the most recent Query, and (b) the feedback returned for that Query.
- Output in the exact format:

```
<interact>
Guess: wA,wB,wC,wD
</interact>
```

- Guess must be comma-separated numbers.

## State Update Rules (in Update rounds, after feedback):
- Each dimension can be adjusted independently, depending on how the feedback affects that hypothesis.

## Query Policy (in Action rounds):
- Ask atomic, clinically meaningful queries that help differentiate among the four hypotheses.
- Avoid repeating previously asked queries.

Before each `<interact>` block, briefly reason (in a few sentences) about what you should do in a: `<scratch>... </scratch>` block, following the protocol above.

Let's get started:

Clinical Vignette: {clinical_vignette}

Medical Question: {medical_question}

Potential Hypotheses: {potential_hypotheses}

Round 1 is an Action Round. Output your first Query.

*Figure 9.* Prompt Template for MediQ.

---

### Input Prompts for the FloDial Dataset

You are an interactive troubleshooting diagnosis agent. Your goal is to iteratively maintain and refine a state vector that tracks the relative plausibility of {num_candidates} candidate descriptions for a given problem, through multi-round interaction.

## Setup:
- You are given: (i) a task description describing the user's problem, and (ii) a list of {num_candidates} candidate descriptions.
- Each candidate is a possible explanation, diagnosis, or resolution suggestion related to the problem.
- Exactly **one** of the candidate descriptions best matches the actual situation described by the user.
- Your goal is to identify which candidate description is the most consistent with the situation, by asking diagnostic yes/no questions.

- Maintain a state vector $(w_1, w_2, \ldots, w_{\{num\_candidates\}})$, where each $w_i \in [0, 1]$, representing your current level of support for candidate $i$.
- Initialization:
    Guess: $0.5, 0.5, \ldots, 0.5$ (length {num_candidates})

- In each Action Round, you issue **only one** query.

## Alternating Interaction Protocol:
Interaction alternates between two types of rounds:

1) **Action Round (odd-numbered rounds: 1,3,5,...)**
- Output only a query in the exact format:

```
<interact>
Query: ...
</interact>
```

- The query must be a single, atomic yes/no diagnostic question intended to reduce uncertainty among the {num_candidates} candidate descriptions.
- Avoid repeating previously asked queries.
- Do **not** directly ask whether a specific candidate description is correct or incorrect.
- Compound or multi-part questions are not allowed.

2) **Update Round (even-numbered rounds: 2,4,6,...)**
- Update and output only the state vector (Guess), based on the most recent query and the feedback returned for that query.
- Output in the exact format:

```
<interact>
Guess: w1,w2,...,w{num_candidates}
</interact>
```

- Each $w_i$ must remain within $[0, 1]$.
- If the user replies "Unknown", leave all weights unchanged.
- Each dimension can be adjusted independently by 0 (not changed), $+0.1$, or $-0.1$, depending on how the feedback affects that candidate.

Before each `<interact>` block, briefly reason (in a few sentences) in a `<scratch>... </scratch>` block about what to ask or how the feedback affects your belief.

Let's get started.

Task Description:
{task_description}

Candidate Descriptions:
{candidate_descriptions}

Round 1 is an Action Round. Output your first query.

*Figure 10.* Prompt Template for FloDial.

