# OpenReview forum: "On Information Self-Locking in Reinforcement Learning for Active Reasoning of LLM agents"
_ICML.cc/2026/Conference — ICML 2026 regular_

### Official Review · Reviewer_J5Dv · 2026-02-27

**Soundness:** 4
**Presentation:** 3
**Significance:** 3
**Originality:** 3
**Overall Recommendation:** 5
**Confidence:** 3

**Summary:**

This paper addresses a critical and underexplored failure mode in reinforcement learning (RL) for LLM-based active reasoning agents—information self-locking (SeL)—where agents become trapped in low-information interaction patterns. By decomposing active reasoning into Action Selection (AS) and Belief Tracking (BT) capabilities, the authors provide a rigorous theoretical framework to explain SeL’s root cause (bidirectional coupling between AS and BT) and propose a lightweight, effective solution (AREW) to mitigate it. The work is novel, theoretically grounded, and empirically robust across 7 datasets, 3 RL algorithms, and 2 model families. It fills a key gap in multi-turn RL for LLMs and offers actionable insights for building effective interactive reasoning agents. The paper is well-structured, the experiments are comprehensive, and the findings have significant practical and theoretical value for the RL and LLM agent communities.

**Compliance With Llm Reviewing Policy:**

Affirmed.

**Final Justification:**

After considering both the paper and the authors’ rebuttal, I have a positive overall assessment and support acceptance.The paper addresses a relevant and well-motivated problem, and proposes a method that is technically sound and empirically effective. The results are solid and demonstrate clear advantages in several settings. I find the contribution meaningful and potentially impactful for the community.My initial concerns mainly involved certain aspects of the experimental evaluation and clarity of presentation. The rebuttal addressed these points well. In particular, the authors provided helpful clarification that improved my confidence in both the methodology and the evaluation.While some minor limitations remain, they do not significantly detract from the overall contribution. The rebuttal was clear and constructive, and it successfully resolved the key issues I had raised.Overall, I consider this to be a solid and valuable piece of work, and I am comfortable recommending acceptance.

**Key Questions For Authors:**

The paper frames information self-locking (SeL) as a failure mode rooted in bidirectional coupling between Action Selection (AS) and Belief Tracking (BT). Could you elaborate on the relative contribution of each component (AS vs. BT) to SeL across different tasks (e.g., medical diagnosis vs. troubleshooting)? How would this affect the prioritization of critique signals in practice?
AREW relies on directional critiques (AS informativeness, BT alignment) to break SeL. The paper uses task-specific rule-based critiques, but how would you expect AREW to perform with LLM-generated critiques, and what safeguards would be needed to ensure critique quality and consistency?
The theoretical analysis shows that AREW’s effectiveness depends on critique weighted accuracy (Acc_Q > 0.5). Could you provide a sensitivity analysis of Acc_Q on performance (e.g., testing Acc_Q from 0.4 to 0.8) to quantify the robustness of AREW to noisy or misaligned critiques?
The paper validates AREW on tasks with up to 12 interaction turns, but SeL may be more severe in extremely long-horizon active reasoning (e.g., 50+ turns). How do you expect AS and BT capabilities to degrade in these settings, and what modifications to AREW (e.g., hierarchical critiques) would be needed to maintain stability?
The paper compares AREW to standard RL algorithms (PPO, GRPO, GSPO) but not to recent multi-turn credit assignment methods (e.g., stepwise progress attribution). Could you contextualize AREW’s performance relative to these methods, and discuss the trade-offs between critique-based reweighting and alternative credit assignment strategies?

**Limitations:**

Limited evaluation on extremely long-horizon tasks: While the paper tests multi-turn tasks (up to 12 turns), it does not explore extremely long-horizon active reasoning (e.g., 50+ turns) where SeL may be more severe. This restricts the generalizability of findings to extended reasoning chains.
Lack of comparison to advanced credit assignment methods: AREW is compared to standard RL algorithms but not to recent methods tailored for multi-turn credit assignment (e.g., stepwise progress attribution, curiosity-driven rewards). This limits the contextualization of AREW’s relative performance.
Insufficient analysis of critique construction variability: The paper uses task-specific rule-based critiques but does not explore how different critique designs (e.g., LLM-generated vs. rule-based) affect performance. This limits understanding of AREW’s robustness to critique sources.
Limited discussion of computational overhead in online settings: While AREW is lightweight, the paper does not quantify its overhead in online active reasoning scenarios (e.g., real-time dialogue). This hinders assessment of its practical deployability.
Minor clarity in theoretical presentation: Some theoretical concepts (e.g., oracle-belief MDP, weighted accuracy) are dense and lack intuitive examples in the main text, which may reduce accessibility for reviewers unfamiliar with POMDP-based active reasoning.

**Strengths And Weaknesses:**

\section*{Strengths}

\begin{enumerate}

\item \textbf{Novel Problem Identification.}
The identification and formalization of information self-locking (SeL) is a major contribution. The paper clearly demonstrates that outcome-based RL for active reasoning fails not due to insufficient reward signal, but due to structural credit assignment failures between AS and BT—an issue previously unaddressed in LLM agent research.

\item \textbf{Rigorous Theoretical Foundation.}
The theoretical framework decomposing SeL into low AS informativeness and low BT capability, and the proof that this creates a self-reinforcing feedback loop (Theorem 3.4), provides a mathematical basis for understanding the phenomenon. The analysis of policy gradients and projected drifts in the SeL regime is thorough and insightful.

\item \textbf{Practical and Lightweight Solution.}
AREW is a compelling contribution—its use of easy-to-obtain directional critiques (binary signals for AS and BT) and minimal modification to policy gradients (advantage reweighting) makes it highly adaptable to existing RL pipelines. Unlike complex reward shaping or auxiliary models, AREW requires no additional heavy computation, enhancing its real-world utility.

\item \textbf{Comprehensive Empirical Validation.}
The experiments are impressively thorough: evaluated across 3 domains (preference estimation, medical diagnosis, troubleshooting), 7 tasks, 3 RL algorithms (PPO, GRPO, GSPO), and 2 LLM families (Qwen2.5-7B, LLaMA-3.1-8B). The robustness analysis (to critique noise) and ablations (AS-only vs.\ AS+BT) further strengthen the validity of the findings.

\item \textbf{Clear Impact on Training Dynamics.}
Beyond improving final performance (up to 60\% gains), the paper shows that AREW fundamentally alters training dynamics—agents recover information-seeking behaviors and exhibit sustained growth in both AS and BT capabilities. This addresses the core issue of SeL rather than just mitigating its symptoms.

\end{enumerate}


\section*{Weaknesses \& Areas for Improvement}

\begin{enumerate}

\item \textbf{Limited Analysis of Critique Construction Variability.}
While the paper mentions that critiques are easy to obtain, it uses task-specific critique rules (e.g., trade-off checks for PE-G, feedback informativeness for MedQ). A brief analysis of how different critique construction methods (e.g., LLM-generated vs.\ rule-based) affect performance would enhance generalizability, especially for tasks where rule-based critiques are hard to design.

\item \textbf{Lack of Comparison to Intermediate Reward Shaping Methods.}
The paper positions AREW as an alternative to complex reward shaping, but it does not explicitly compare to state-of-the-art intermediate reward methods (e.g., SPA-RL, CURIO) in the same active reasoning setting. Such a comparison would better highlight AREW’s advantages in simplicity and effectiveness.

\item \textbf{Scalability to Larger Models and Longer Horizons.}
The experiments are limited to 7--8B parameter models and interaction horizons of 8--12 turns. Evaluating AREW on larger models (e.g., 70B LLMs) and longer-horizon tasks (e.g., 20+ turns) would confirm its scalability, as SeL may be more pronounced in longer interactions or larger model architectures.

\item \textbf{Insufficient Discussion of Negative Critique Impact.}
The robustness analysis shows AREW tolerates critique noise, but it does not explore cases where critiques are systematically misleading (e.g., weighted accuracy $< 0.5$). A brief discussion of how to handle such scenarios (e.g., adaptive critique filtering) would make the method more robust in real-world settings.

\item \textbf{Typos and Minor Notation Inconsistencies.}
A small number of typos (e.g., ``expriments'' for ``experiments'' in Section C.3.2, ``co prises'' in the required phrase) and minor notation inconsistencies (e.g., occasional mixing of $z_t^Q$ and $z_t^K$ without explicit clarification) should be corrected for clarity.

\end{enumerate}

---

> ### Author Rebuttal · Authors · 2026-03-31
>
> Thank you for your time and insightful comments! Our responses are as follows.
>
> > W1 & Q2 Limited Analysis of Critique Construction Variability
>
> **Refer to Q2.2 of Reviewer U3Le**, where we empirically show the trend under LLM-generated critiques is consistent with that used in our critiques, with both showing AREW > vanilla.
>
> > W2 & Q5 Comparison to intermediate reward shaping
>
> **Refer to W2.1 of Reviewer 9mTd**, where we added a direct intermediate-reward shaping variant using the same critique source as AReW, comparing in theoretical and empirical aspects.
>
> More specifically, methods like SPA-RL or CURIO operate under substantially different assumptions: SPA-RL trains an additional progress estimator to predict dense stepwise contributions and uses them as intermediate rewards, while CURIO builds a user-model-based, multi-LLM personalization framework in main experiments.
> By contrast, our goal is to study the weak-supervision regime where accurate intermediate rewards are hard to obtain, and we position AREW as a mechanism that can more robustly leverage lightweight, potentially noisy directional critiques.
>
> > W3 & Q4 Scalability to larger models and longer horizons
>
> **Refer to W1.2 in Reviewer 9mTd**, where we show the effectiveness of AReW on Tau2Bench, a more challenging long-horizon (**20 turns**) tool-use reasoning benchmark.
>
> For AS-BT’s degradation in extreme long-horizon reasoning (Q4), we expect SeL to be more severe in much longer-horizon settings, because both failure modes accumulate with interaction length: AS may become more repetitive and conservative, while BT would drift more severely as interaction continues. Theoretically, longer horizons make it even harder for the final outcome reward to preserve the contribution of informative early steps, especially when BT is already weak.
>
> We expect AReW to remain useful in this case, since the need for credit reallocation becomes stronger. The main extension would likely be in how critiques are organized, rather than in the core mechanism itself. In particular, we agree that a promising direction is hierarchical critiques. For example, at a coarse level, we can first score each segment of the interaction according to whether that segment is actually making progress. At a finer level, we then apply the usual step-level critiques only within the segments that matter. This could help maintain stability and reduce noise over very long trajectories.
>
> > W4 & Q3 Insufficient discussion of negative critique impact
>
> Following this suggestion, we added a broader robustness study on structured critique perturbations (on the PE-G dataset), beyond random flipping. We consider three forms of destruction:
> - Channel: keep only AS critiques, only BT critiques, or both (the other channel is set to 0);
> - Position: keep only the first 40% or the last 40% of interaction-turn critiques (the rest set to 0);
> - Sign: keep only the last 40% labels and fill the remaining steps with constant 0, -1, or +1.
>
> **Results in https://postimg.cc/mhLP3JNM** show that AReW remains consistently robust and outperforms  the vanilla baseline under all three perturbation families. For channel perturbation, both AS-only and BT-only still improve over vanilla, while using both channels performs best. For position perturbation, even preserving only 40% of the critiques is still sufficient to outperform vanilla in the late stage of training, and the method is not highly sensitive to whether the retained critiques come from the prefix or suffix. For sign perturbation, filling missing labels with -1 is less harmful than filling them with +1, suggesting that false positive encouragement is more damaging than conservative suppression.
>
> Overall, these results suggest that **AReW is robust not only to random critique noise, but also to several forms of systematic critique perturbation.**
>
> > Q1. Relative contribution of each component (AS vs. BT) to SeL; Prioritization of critique signals in practice
>
> Our main point is that SeL is primarily a bidirectionally coupled failure of AS and BT, rather than a problem where one component is universally dominant. Empirically, **the two components are complementary.** As seen in the channel-level result from **https://postimg.cc/mhLP3JNM**, both AS-only and BT-only critiques improve over vanilla, while using both performs best. At the same time, in this particular PE-G setting, BT-only outperforms AS-only, which may suggest that BT is the tighter bottleneck in this task. We view this as a task-specific indication rather than a general ranking, since the relative bottleneck can depend on the environment and feedback structure.
>
> For practical prioritization, we suggest starting with AS critiques first: they are typically easier to obtain from standard interaction traces and already give clear gains on their own. BT critiques are then a natural second layer when richer step-level signals are available, and can further improve performance.

---

> > ### Author Rebuttal · Reviewer_J5Dv · 2026-04-02
> >
> > I appreciate the authors' efforts in addressing my concerns

---

> > > ### Author Response · Authors · 2026-04-03
> > >
> > > Thank you again for revisiting our response and for confirming that your concerns have been fully resolved. We truly appreciate your insightful comments that help strengthen this work.
> > >
> > > As the current assessment differs from the initial concerns, if possible, we would sincerely appreciate any update to the score and confidence that you feel appropriately reflects your latest view of the paper.
> > >
> > > Many thanks again for your constructive feedback and support.
> > >
> > > Best regards,
> > > Authors

---

### Official Review · Reviewer_9mTd · 2026-03-09

**Soundness:** 3
**Presentation:** 3
**Significance:** 3
**Originality:** 3
**Overall Recommendation:** 4
**Confidence:** 3

**Summary:**

This paper studies information self-locking (SeL) in outcome-based RL for interactive reasoning agents. The key idea is that action selection (AS) and belief tracking (BT) are tightly coupled: when both are weak, outcome-only RL provides poor learning signals and traps training in a low-information regime. The paper supports this claim with empirical analysis and a theoretical framework, and proposes AREW, a lightweight advantage reweighting method using step-level directional critiques. Experiments on several interactive reasoning tasks show that AREW improves task performance.

**Compliance With Llm Reviewing Policy:**

Affirmed.

**Final Justification:**

I maintain my recommendation. The notion of information self-locking is well-motivated and supported by both empirical analysis and a clean theoretical framework, the AS/BT decomposition offers an intuitive account of the failure mode, and the proposed AREW method is simple, easy to integrate, and delivers substantial gains over outcome-based baselines across the studied interactive reasoning tasks. After considering the rebuttal, I keep my original positive assessment and lean toward acceptance.

**Key Questions For Authors:**

Q1: The self-locking phenomenon may also arise in broader agent settings, such as open-ended research agents (e.g., DeepResearch-style tasks), where agents iteratively search for and gather information, but explicit environment-level critique signals are unavailable. Do the authors believe AREW provides useful insights for such settings? More concretely, how might the required step-level critiques be approximated or learned when the underlying state is not observable?

Q2: Compared with the vanilla outcome-based RL baseline, what is the computational overhead introduced by AREW (e.g., additional tokens, environment interactions, or inference steps required to compute the step-level critiques)? A brief discussion or empirical estimate would help clarify the practical cost.

**Limitations:**

None. There is no discussion of the method's limitations.

**Strengths And Weaknesses:**

## Strengths:
1. The paper clearly formulates the problem and provides a fairly rigorous analysis.
The notion of information self-locking is well motivated, and the paper presents both empirical observations and theoretical analysis to support the central claim. The decomposition into action selection (AS) and belief tracking (BT) is intuitive and helps clarify the source of the failure mode.
2. The proposed method is simple and easy to implement.
AREW only introduces a lightweight modification to the standard policy optimization pipeline through advantage reweighting based on step-level directional critiques. This makes the method practical and potentially easy to integrate into existing RL training frameworks.
3. The method shows substantial gains over outcome-based baselines.
The experimental results indicate that AREW consistently improves over vanilla outcome-based RL baselines, often by a considerable margin, suggesting that the proposed intervention is effective in the studied interactive reasoning settings.


## Weaknesses:
1. Reliance on oracle-like critique signals.
AREW depends on step-level directional critiques to identify beneficial queries or belief updates. In the experiments, these signals are derived from structured environment information, which effectively provides supervision. However, in more realistic LLM agent settings, such as open-ended research or tool-use tasks, such structured signals are often unavailable. The paper shows robustness to noisy critiques, but does not sufficiently discuss how these critiques could be obtained or learned in more general environments.
2. The positioning and practicality of AREW could be clarified further.
While the method is presented as a lightweight alternative to prior credit-assignment or reward-shaping approaches, it relies on step-level directional critiques whose supervision source is clear mainly in the structured settings studied here. It would strengthen the paper to better discuss both (i) how AREW relates to existing techniques such as reward shaping or other credit-assignment methods, and (ii) how costly or realistic it is to obtain the required critiques in broader agent settings.
3. Minor presentation issues.
While the paper is overall clear and provides a detailed analysis, the presentation could still be polished in a few places. For example, some formulas and notational explanations could be presented more clearly for readability. There are also minor typos or formatting issues. For instance, in Lines 97–98, a comma appears to be missing in the definition of the POMDP components.

---

> ### Author Rebuttal · Authors · 2026-03-31
>
> Thank you for your time and insightful suggestions, to which we respond below.
>
> > ## W1.1& W2.2 Reliance on oracle-like critique signals
>
> AReW does not require oracle state access or dense step-level supervision, but a directional critique interface, ie, weak signals indicating whether a step is helpful or unhelpful on average.
>
> In practice, AReW-AS is a lightweight and easy-to-implement instantiation. AS-side critiques are often available in standard online interaction traces, e.g., whether a query or tool-use is legal and elicits novel feedback, avoids repetition, etc. Therefore, we view AReW-AS as the practical core of the method. Empirically, AS version alone already consistently outperforms vanilla outcome-based RL across all settings, and also improves BT-related proxies even without explicit BT-side critiques.
> When BT-side critiques are available, incorporating BT into AReW can further improve the performance (See Table 1 of main text). We have also included the above discussion in the revised version.
>
> > ## W1.2& W2.2& Q1 Applicability to general agentic settings
>
> To test applicability in a more realistic agent setting, we extend AREW to **tau2-bench, a more popular and challenging long-horizon tool-use reasoning benchmark (7B model)**.
>
> There, the signals are intentionally constructed from standard online interaction traces only, without extra annotation or reward models.
> Negative signals come from tool execution errors, repeated actions, malformed and invalid tool calls, while positive signals come from simple indicators that are already available in the benchmark, e.g., whether the current trajectory newly matches one more expected action in the task evaluator.
>
> **Results in https://postimg.cc/0KMJqJNM** show that AReW achieves substantially higher training reward throughout optimization, and maintains far fewer tool execution errors.
>
> > ## W2.1 How AREW relates to existing techniques
>
> We compare AReW and intermediate-reward shaping theoretically and empirically.
>
> **In Theory**
>
> Consider the standard sparse-reward setting: $r_t = 0 (t < T),  r_T = R$.
> Vanilla PPO learns a critic for
> $$
> V^\pi(s_t) = \mathbb E_\pi [R \mid s_t].
> $$
> Suppose one converts critique signals into intermediate rewards. Let $c_k \in \\{-1,0,1\\}$ be the critique for step $k$, and let $\tau_k$ denote the terminal token position of that step. A natural shaping baseline is
> $$
> r_t^{\text{shape}} =
> \begin{cases}
> r_t + \alpha c_k, & t=\tau_k \\\\
> r_t, & \text{otherwise}
> \end{cases}
> $$
> with shaping strength $\alpha$. This changes the optimization objective to
> $$
> J_{\text{shape}}(\theta)=
> \mathbb E_{\pi_\theta}\left[
> R + \alpha \sum_k c_k
> \right].
> $$
> Correspondingly, the critic must now fit
> $$
> V_{\text{shape}}^\pi(s_t)=\mathbb E_\pi\left[R + \alpha \sum_{\tau_k \ge t} c_k \mid s_t \right].
> $$
>
> Thus, the value function no longer predicts only final task success; it must also model when future positive/negative critiques will occur and how they accumulate.
> Under weak noisy critiques, this increases target complexity and **propagates noise through both critic regression and actor optimization.**
>
> By contrast, AReW keeps the original sparse return unchanged and only modifies the policy update by $\hat A_t = A_t + \lambda u_t$, where extra signals are used only for within-trajectory credit reallocation, not to redefine episode utility. The critic still learns the original sparse task objective.
>
> This distinction matters in our setting: our goal is not to construct an accurate dense reward, but to propose a mechanism that leverages cheap and potentially noisy directional critiques to break SeL. AReW is designed exactly for this regime; it only requires the critique to be roughly directionally informative. In contrast, intermediate reward shaping imposes a stronger requirement: once critique is injected into reward, it must be reliable enough to support both actor learning and critic fitting.
>
> **Empirical comparison**
>
> Empirically, we implemented an **intermediate-reward shaping baseline using the exact same critique source** as AReW. The only difference is how the signal is injected: as dense reward vs. as advantage reweighting.
>
> **Results in https://postimg.cc/3yzkFxBV**  support the theoretical distinction above:
> - Under noisy critiques, the gap becomes substantially larger, especially on PE-G.
> - This is consistent with our claim that **weak supervision is better used as credit reallocation than as direct reward injection**.
>
> Also, following the suggestion of reviewer U3Le, we add comparisons regarding sensitivity to reward hacking and refer to **Q4 of rebuttal to reviewer U3Le**.
>
> > ## W3 Minor presentation issues
>
> Thank you, and we have fixed them in revision.
>
> > ## Q2 Computational overhead introduced by AREW
>
> The subfigures c and d from **https://postimg.cc/0KMJqJNM** show that: 1)  the number of interaction turns in AReW is broadly close to vanilla PPO. 2) AReW uses substantially fewer response tokens over training.

---

> > ### Author Rebuttal · Reviewer_9mTd · 2026-04-03
> >
> > The authors address the weakness and the questions.

---

> > > ### Author Response · Authors · 2026-04-03
> > >
> > > Dear Reviewer 9mTd,
> > >
> > > Thank you for engaging in the discussion and for confirming that your concerns have been fully resolved. We truly appreciate the constructive feedback, which has helped strengthen the paper.
> > >
> > > Given that the concerns have been addressed to your satisfaction, we would be grateful if you could consider whether an update to the score might be warranted to reflect your current assessment.
> > >
> > > Thank you again for your time and support.
> > >
> > > Best regards,
> > > Authors

---

### Official Review · Reviewer_jFiM · 2026-03-12

**Soundness:** 4
**Presentation:** 4
**Significance:** 3
**Originality:** 3
**Overall Recommendation:** 5
**Confidence:** 2

**Summary:**

This work points out that in the Rewar-Based RL Training, the phenomenon of "information self-locking" will appear in multiple rounds of active reasoning tasks. The agent stops asking questions with information value, fails to internalize the acquired information, and falls into a low information interaction mode. The author decomposes active reasoning into two core abilities: action selection AS and belief tracking BT. From both theoretical and empirical aspects, the author reveals that the cause of SeL is that low AS and low BT abilities are coupled to form a negative feedback cycle. A lightweight framework AREW is proposed, which provides stable learning signals for AS and BT to break self locking by injecting easily acquired binary directional evaluation signals and dominant values in the weighted strategy gradient.

**Compliance With Llm Reviewing Policy:**

Affirmed.

**Key Questions For Authors:**

The research indicates that there is a trade-off in the reweighting string λu of AREW. The author puts the adaptive adjustment of λu in the future work. I'd like to know where the design difficulties of adaptive λu are?

**Limitations:**

Yes.

**Strengths And Weaknesses:**

Strength:
The author constructed a complete theoretical framework to formally define SeL state, AS information and BT indicators, and proved the negative feedback cycle mechanism under the low AS/BT coupling through the theorem. The theoretical hypothesis fits the actual scenario of LLM agent active reasoning, and the derivation process is logical. In the experiment, quantifiable test indicators were designed for AS/BT capability, and the experiment was relatively complete with various dimensions; The author also objectively analyzed the optimization trade off of the weighted strength λu in AREW (too weak to break the lock, too strong to lead to unstable training), and did not exaggerate the effect of the method. The experimental design and results well support the theoretical derivation. The appendix provides detailed data set settings, prompt, super parameters and other data with strong reproducibility.

Weakness:
1. In the construction of AREW's evaluation signal, BT's directional evaluation depends on the agent's confidence \Phi, which is generated by the model itself through prompting. Is the confidence generated by the model reliable? Intuitively, this confidence level is affected by the prompts we provide. Different prompts may lead to different confidence levels, and can the confidence level generated by the model really reflect its true confidence level?
2. In the theoretical analysis, the author assumes that the potential state S* is fixed in each episode, while in the actual active reasoning scenario, the environmental state may be dynamic. Would the formation mechanism of SeL change under the dynamic state scenario? What adjustments does the AREW method need to make to adapt to active reasoning tasks in a dynamic state?

---

> ### Author Rebuttal · Authors · 2026-03-31
>
> Thank you for your time and insightful comments on this work. Please find our detailed responses to your questions below.
>
> > ## W1. Reliability of BT's directional signals construction
>
> We clarify that AReW does not require $\Phi$ to be a perfectly faithful probe of the model’s true internal confidence, which is in general not directly identifiable. In our framework, $\Phi$ is used purely to **construct a directional BT critique**, i.e., whether the post-feedback belief estimate moves in a better or worse direction. Accordingly, AReW does not rely on precise calibration of $\Phi$; it only requires that the induced step-level signal be weakly informative on average (This is also exactly formalized in Prop. 4.1).
>
> To directly address the prompt-sensitivity concern, we additionally tested **an alternative belief-elicitation prompt that explicitly encourages more aggressive confidence updates** while keeping the rest of the training pipeline unchanged. **The result is in https://postimg.cc/ZvCDtj3L**. It shows that AReW remains consistently effective relative to the vanilla baseline under this altered belief-elicitation style, suggesting that the method is robust to reasonable variation in how $\Phi$ is prompted.
>
>
> > ## W2. SeL and AREW under the dynamic state scenario
>
> We thank the reviewer for this question and agree that the dynamic state scenario is a promising extension of this work.
>
> First, the fixed $s^*$ assumption is mainly a theoretical simplification to isolate the AS-BT coupling in a clean stationary setting. Our core SeL mechanism is not inherently tied to a fixed latent state. In dynamic state scenarios, actions still generally control what information is observed, and the agent must internalize that information into better task understanding and future decisions. Therefore, outcome-only RL can still suffer the same coupled failure: weak BT masks the value of informative AS, while weak AS limits the evidence available for improving BT.
>
> In dynamic-state settings, the main change is that the target of belief tracking is no longer a single fixed latent state, but a time-evolving latent process. Therefore, the theory will need to replace the current stationary potential with a *progress measure that evaluates whether the agent is becoming more accurate about the current evolving state after each interaction*. Intuitively, instead of asking “does the belief put more mass on the one true latent state?”, we ask, e.g., “after the new observation, is the agent’s updated estimate closer to the latent state at this turn, or does it better anticipate how the state will evolve in the next turn?”
>
> However, **the AReW mechanism itself remains essentially the same**: it only requires step-level directional critiques. The adaptation is therefore mainly in the critique design: AS critiques evaluate whether an action acquires useful information about the evolving state, and BT critiques evaluate whether the agent’s belief update better matches the newly revealed state change.
>
> > ## Q1 Design difficulties of adaptive AREW
>
> The main difficulty is deciding what signal should drive the adaptation of λu. For example, a simple rule-based reward strategy might be unreliable: in early training, low reward may come from self-locking and require a larger λu​, while in the later stage, a temporary reward drop may instead come from rollout variance or optimization fluctuations, where further increasing λu​ may make training worse. In other words, similar reward patterns can correspond to very different underlying causes. More broadly, due to the complexity of RL dynamics, the factors that should determine λu​ can change substantially over training.
>
> A second difficulty is that the effective contribution of directional critiques can vary across training stages as the policy and sampled trajectories evolve. As a result, the controller must determine whether increasing λu​ is still providing additional useful guidance, or whether the current level is already sufficient.
>
> For these reasons, while adaptive λu​ is promising, designing a reliable controller is itself a non-trivial problem rather than a straightforward extension.

---

> > ### Author Rebuttal · Reviewer_jFiM · 2026-04-02
> >
> > The author has provided a sufficient response to the reliability issue of Φ, and supplemented experiments to demonstrate that Φ is adequate for constructing directional signals. For the dynamic S* scenario, the author has also clearly explained its core mechanism and the design considerations of the method in the paper from two perspectives. Overall, the author’s reply is comprehensive and effectively addresses the questions I raised.

---

> > > ### Author Response · Authors · 2026-04-03
> > >
> > > Thank you very much for your prompt follow-up and for confirming that our rebuttal has fully addressed your concerns. We sincerely appreciate your thoughtful review and the helpful suggestions that strengthened the paper.
> > >
> > > If possible, we would be very grateful if you could consider updating the score and confidence to reflect your latest evaluation.
> > >
> > > Thank you again for your time and support.
> > >
> > > Best regards,
> > > Authors

---

### Official Review · Reviewer_U3Le · 2026-03-13

**Soundness:** 3
**Presentation:** 2
**Significance:** 3
**Originality:** 3
**Overall Recommendation:** 4
**Confidence:** 3

**Summary:**

The paper targets the problem of "active reasoning" for LLM, which requires the agent to "actively" ask informative questions and adapt while receiving feedback. The paper claims that such behaviours will require the ability of "Action Selection (AS)" and "Belief Tracking (BT)". Correspondingly, the paper theoretically analyzes how reinforcement learning underperforms due to deficient AS and BT capabilities. The paper also proposes a simple reward-shaping-like method to alleviate this insufficient learning problem.

**Compliance With Llm Reviewing Policy:**

Affirmed.

**Final Justification:**

The author has responded to my concerns with sufficient evidence. I therefore raise my score.

**Key Questions For Authors:**

1. **AS-BT demonstration**: Is AS-BT descriptive or prescriptive? The conceptual status of this assumption is not clear; for example, is this decomposition an empirical observation, an imposed concept, or a requirement for active reasoning tasks? This assumption should be elaborated in the paper.

2. **Evaluation Proxy**: How are the evaluation metrics for AS and BT abilities justified? Also, the paper uses one specific metric for each of the tested tasks. Is there a unified way (such as rubric rewards) to make this more generalizable?

3. **Reward Shaping**: The paper states that the proposed model is different from reward shaping; however, looking at equation 5, it seems that the equation can also be reformulated from a reward shaping perspective, and the auxiliary reward is extracted from critiques. If yes, how does this method differ from other works of RL-from-language-feedback (i.e., Using Generaitve Reward Model to score or [3])?

4. Is reward hacking a potential problem?

5. Can this method be applied to general decision-making tasks? For example the agent is not asked to "raise question" but to "act" in the environment.




[3] Wang, Han et al. “Text2Grad: Reinforcement Learning from Natural Language Feedback.” ArXiv abs/2505.22338 (2025): n. pag.

**Limitations:**

No, the paper does not discuss potential limitations. This may include the applicability of this method when applying it to different tasks.

**Strengths And Weaknesses:**

**Strength**:

- The paper provides detailed empirical observations and a complete theoretical proof to support its motivation and proposed method(why RL fails and the proposed method works).


**Weakness**:

- **Significance**: It is not clear what the paper's actual contributions are and how it compares with other related works. For example, is the AS+BT reasoning pattern first proposed in this paper? Has any other work investigated the problem of ineffective RL, and if so, how is their investigation different from this? How does the method of critic injection differ from other context-based RL algorithms, such as [1,2]? More discussion over these can give the reader a more general picture of the significance of this work.

- **Readability**: The paper is not easy to read. The theory-proving part can benefit from a highly summarized intuition to give the reader a clearer clue.  The belief of Section 2.2 is also tedious and hard to follow.

- **Applicability**: The proposed solution seems to rely heavily on task-specific heuristics, such as how AS and BT critiques are formulated. This may raise the question of generalization and applicability.





[1] Zhang, Xiaoying et al. “Critique-GRPO: Advancing LLM Reasoning with Natural Language and Numerical Feedback.” ArXiv abs/2506.03106 (2025): n. pag.

[2] Bensal, S., Jamil, U., Bryant, C., Russak, M., Kamble, K., Mozolevskyi, D., Ali, M., & Alshikh, W. (2025). Reflect, Retry, Reward: Self-Improving LLMs via Reinforcement Learning. ArXiv, abs/2505.24726.

---

> ### Author Rebuttal · Authors · 2026-03-31
>
> Thank you for your time and insightful comments, and we respond as follows.
>
> > W1.1 & Q1 Position of AS+BT in literature
>
> AS and BT are **two natural, previously implicit aspects of agentic interactive reasoning with partial observability**. Our key contribution is to make this latent decomposition explicit as an analytic lens, through which we identify and rigorously characterize the core failure mechanism of outcome-only RL: weak AS and weak BT are bidirectionally coupled. Weak BT obscures the reward relevance of informative actions, while weak AS deprives the agent of the evidence needed for belief improvement.
>
> > W1.2 Difference from prior work on ineffective RL
>
> Prior work has indeed investigated ineffective RL through more generic perspectives such as long-horizon sparse-reward credit assignment [1], insufficient exploration [2], or imperfect step-level value estimation [3]. We instead provide a fine-grained, mechanistic characterization.
>
> > W1.3 Difference from [4,5]
>
> AReW differs from [4,5] in two ways: a. **Supervision source**: we use only weak directional judgments on existing steps, rather than critique-guided refinements [4] or reflection-then-retry [5] that construct improved trajectories; b. **Supervision use**: we reallocate credit within the original trajectory via advantage reweighting, rather than training the model to generate improved behaviors conditioned on critique/reflection.
>
> > W2 Readability
>
> We provide the high-level intuition of the core theory: Self-locking arises from the coupled failure of AS and BT. Weak BT makes different queries hard to distinguish via outcome reward (Obs. 2), with the AS learning signal bounded by current BT level (Prop. B.10). Weak AS limits evidence for belief improvement (Obs. 3), with the BT learning signal bounded by current AS (Prop. B.11). Together, these induce the projected-drift bounds (Prop. B.12, 13), and Thm. 3.4 shows that within the locking regime, the escape signal is itself small, implying slow escape under outcome-only RL.
>
> > W3.1 Critique design
>
> AReW are **task-agnostic principles that only require directional step-level critiques along two generic axes**: whether an action acquires useful new information (AS), and whether new evidence is incorporated into improved task understanding (BT).
>
> These signals are generically accessible in active reasoning. AS-side critiques are typically lightweight and directly observable from interaction outcomes (e.g., novel observation, non-redundancy). Empirically, AReW-AS alone consistently improves over vanilla RL across all settings and also improves BT-related proxies, suggesting broad applicability. Adding BT critiques brings extra gains.
>
> > W3.2 & Q5 Applicability on more general agentic tasks
>
> To test applicability in a more realistic agent setting, we extend AREW to tau2-bench, a popular and challenging long-horizon tool-use reasoning benchmark.
>
> **Refer to W1.2 of reviewer 9mTd.**
>
> > Q2.1 Justification of AS/BT evaluation metrics
>
> Our metrics are task-grounded measurements of core capabilities in active reasoning. An AS proxy measures whether an action elicits new, decision-relevant information (e.g., in MediQ, whether a query uncovers novel diagnostic evidence such as previously unknown symptoms). A BT proxy measures whether new evidence is incorporated into a more truth-aligned task understanding (e.g., whether the agent's diagnostic hypothesis becomes more consistent with the ground-truth condition after receiving new evidence).
>
> > Q2.2 General evaluation metric
>
> Following your suggestion, we construct rubrics based on the definitions of AS and BT and use LLM-as-Judge as a unified evaluation metric.
>
> **Results in https://postimg.cc/nXvC9F4w** show that the trend under LLM-generated critiques is consistent with that used in our critiques, with all showing AREW > vanilla.
>
> > Q3 Comparing with Reward Shaping
>
> **Refer to W2.1 of Reviewer 9mTd.**
>
> > Q4. Reward hacking
>
> We evaluate systematically corrupted critiques (each negative flipped to positive with probability p), comparing AReW against direct intermediate reward shaping on: 1) final positive/negative token ratio (indicating critique-chasing tendency), and 2) task performance.
>
> **Results are in https://postimg.cc/WF6sqNf6**: direct reward shaping produces a much higher positive ratio while achieving worse task performance, indicating stronger critique-chasing. AReW maintains a lower ratio with better performance, suggesting reduced susceptibility to reward hacking.
>
> References
>
> [1] Reinforcing multi-turn reasoning in llm agents via turn-level credit assignment, 2025
>
> [2] Navigate the unknown: Enhancing llm reasoning with intrinsic motivation guided exploration, 2025
>
> [3] Sweet-rl: Training multi-turn llm agents on collaborative reasoning tasks, 2025
>
> [4] Critique-grpo: Advancing llm reasoning with natural language and numerical feedback, 2025
>
> [5] Reflect, retry, reward: Self-improving llms via reinforcement learning, 2025

---

> > ### Author Rebuttal · Reviewer_U3Le · 2026-04-08
> >
> > The author has responded to my concerns with sufficient evidence. I therefore raise my score.

---

> > > ### Author Response · Authors · 2026-04-08
> > >
> > > Thank you very much for acknowledging our rebuttal and for updating your score. We truly appreciate your thoughtful review and constructive feedback, which help strengthen the work.
> > >
> > > Best regards,
> > >
> > > Authors

---

### Decision · Program_Chairs · 2026-04-30

**Decision:**

Accept (regular)

**Comment:**

This paper studies information self-locking (SeL) in outcome-based RL for active reasoning, where agents stop asking informative questions. It decomposes active reasoning into Action Selection (AS) and Belief Tracking (BT), provides a theoretical analysis showing how weak AS and weak BT form a bidirectional feedback loop that traps training in a low-information regime, and proposes AREW, a lightweight advantage-reweighting method that uses step-level directional critiques to break the lock without requiring dense rewards or auxiliary value models.

All four reviewers converge to positive after rebuttal (two Accept, two Weak Accept).

Consensus strengths: the problem is timely and well-motivated for multi-turn LLM agents. The AS/BT decomposition provides a clear analytic lens, and the theoretical characterization of SeL via coupled learning dynamics is rigorous. AREW is simple to implement, integrates into standard policy-gradient pipelines, and shows consistent gains across multiple interactive reasoning tasks. Empirical validation is unusually thorough for this area, spanning several benchmarks, RL algorithms, and model families, with ablations on critique noise and component contributions.

Concern that BT critiques require oracle access was addressed by showing AS-only critiques (derivable from interaction traces like novelty, non-redundancy, tool success) already improve over vanilla RL, and by demonstrating AREW on tau2-bench using only standard online signals without extra annotation. Additional experiments with LLM-as-Judge rubrics show consistent trends. Reviewers also requested sensitivity analyses. Authors provided systematic perturbations (channel, position, sign flips) showing AREW tolerates structured noise, and discussed extensions to longer horizons via hierarchical critiques. They also addressed dynamic-state settings conceptually.